# A regulatory network of Sox and Six transcription factors initiate a cell fate transformation during hearing regeneration in adult zebrafish

## Graphical abstract

## Authors

Erin Jimenez, Claire C. Slevin,
Wei Song, ..., Abdel G. Elkahloun,
Ivan Ovcharenko, Shawn M. Burgess

## Correspondence

burgess@mail.nih.gov

## In brief

Jimenez et al. interrogate the epigenomic and transcriptomic landscape of regenerating adult zebrafish inner-ear sensory epithelia. They show that the support-cell population transitions to an intermediate "progenitor" cell state that becomes new hair cells, and they demonstrate that the cell fate decisions may be driven by the coordinate regulation and spatial co-binding of Sox and Six transcription factors. By functionally validating a predicted regeneration-responsive enhancer upstream of *sox2*, they show that precise timing of *sox2* expression is critical for hearing regeneration in zebrafish.

## Highlights

- Integrated scRNA-seq and scATAC-seq of zebrafish inner ears during hearing regeneration

- Sox transcription factors trigger the regeneration response in the support cells

- Sox and Six factors cooperate spatially during hair cell differentiation

- An essential enhancer controls the precise timing of *sox2* expression during regeneration

Jimenez et al., 2022, Cell Genomics 2, 100170
September 14, 2022

CellPress

## Article

# A regulatory network of Sox and Six transcription factors initiate a cell fate transformation during hearing regeneration in adult zebrafish

Erin Jimenez,[1] Claire C. Slevin,[1] Wei Song,[4] Zelin Chen,[2,3] Stephen C. Frederickson,[1] Derek Gildea,[1] Weiwei Wu,[5] Abdel G. Elkahloun,[6] Ivan Ovcharenko,[4] and Shawn M. Burgess[1,7,*]

[1]Translational and Functional Genomics Branch, National Human Genome Research Institute, Bethesda, MD, USA
[2]CAS Key Laboratory of Tropical Marine Bio-Resources and Ecology, South China Sea Institute of Oceanology, Chinese Academy of Sciences, Guangzhou, China
[3]Southern Marine Science and Engineering Guangdong Laboratory, Guangzhou, China
[4]Computational Biology Branch, National Center for Biotechnology Information, National Library of Medicine, National Institutes of Health, Bethesda, MD 20894, USA
[5]Vaccine Immunology Program, Vaccine Research Center, National Institute of Allergy and Infectious Diseases, Bethesda, MD, USA
[6]Cancer Genetics and Comparative Genomics Branch, National Human Genome Research Institute, Bethesda, MD, USA
[7]Lead contact
*Correspondence: burgess@mail.nih.gov

## SUMMARY

Using adult zebrafish inner ears as a model for sensorineural regeneration, we ablated the mechanosensory receptors and characterized the single-cell epigenome and transcriptome at consecutive time points during hair cell regeneration. We utilized deep learning on the regeneration-induced open chromatin sequences and identified cell-specific transcription factor (TF) motif patterns. Enhancer activity correlated with gene expression and identified potential gene regulatory networks. A pattern of overlapping Sox- and Six-family TF gene expression and binding motifs was detected, suggesting a combinatorial program of TFs driving regeneration and cell identity. Pseudotime analysis of single-cell transcriptomic data suggested that support cells within the sensory epithelium changed cell identity to a "progenitor" cell population that could differentiate into hair cells. We identified a 2.6 kb DNA enhancer upstream of the *sox2* promoter that, when deleted, showed a dominant phenotype that resulted in a hair-cell-regeneration-specific deficit in both the lateral line and adult inner ear.

## INTRODUCTION

The capacity to regenerate tissues after injury unevenly manifests across the vertebrate lineage.[1] In most mammals, consistent cellular renewal is limited to certain cell types, such as skin, gut, and blood, while major tissue regeneration is even further restricted to a small number of organs, such as the liver. Damage to the mammalian inner ear sensory epithelium is irreversible and results in permanent hearing loss or vestibular defects. Interestingly, this is a feature that sets mammals apart from most other vertebrates who can continually produce new hair cells (HCs) throughout their lifetimes and/or can regenerate them in response to trauma.

The HCs are mechanosensory receptors used in the inner ear auditory and vestibular organs of all vertebrates and in the lateral line systems of aquatic vertebrates.[2] In fish these organs are the saccule and utricle, respectively (Figure S1A).[3] The saccule in fish primarily detects acoustic vibrations (amplified through the body instead of via an eardrum), while the utricle primarily functions as a gravitation sensor but has also shown some auditory

potential.[2,4] HCs similar to those that reside in the inner ear are also located on the skin in fish and amphibians in small structures called "neuromasts," which reside in an organ normally referred to as the "lateral line."[2,3] Although lateral line HCs differ in morphology compared with the adult inner ear, the accessibility of the lateral line HCs on the skin surface has made them a popular *in vivo* model. However, more effort is justified in studying inner ear HC regeneration in fishes with the goal of restoring lost hearing in mammals.

Key genes in inner ear development can also have important roles in regeneration.[5] However regeneration is distinct from embryonic development[6] and recent genome-wide analyses suggest that, while regeneration programs may target many of the same genes, they may do so through distinct regulatory sequences:[7,8] reviewed in Rodriguez et al.[9] and Yang et al.[10]

Enhancer elements are critical in the control of development,[11] and several groups have made connections between enhancer regulation and tissue regeneration programs. Injury-responsive or regeneration-associated enhancers that direct gene expression in injured tissues have been identified in the regenerating

heart and fin of zebrafish.[7,9,12–20] Comparative epigenomic profiling and single-cell genomics experiments have revealed species-specific and evolutionarily conserved genomic responses to regeneration in fish termed tissue regeneration enhancer elements (or TREEs).[7,13,21] However, ablation of enhancers in these studies showed that, even though these enhancers respond to injury, they are generally not essential for normal regeneration.

Here, we profiled changes in chromatin accessibility (scATAC-seq) and gene expression single-cell 3′ RNA sequencing (scRNA-seq) during regeneration of the zebrafish inner ear at single-cell resolution. We showed the support cells (SCs) potentially transitioned into a "progenitor-like" state that differentiated into new HCs. We also identified a key regulator of *sox2* expression that, when deleted, the HCs developed normally, but HC regeneration after injury was significantly disrupted.

## RESULTS

### scRNA-seq identifies different cell populations in the inner ear

We used Tg(*myo6b*:hDTR) transgenic zebrafish which permits conditional and selective ablation of HCs in the adult zebrafish inner ear.[22] The Tg(*myo6b*:hDTR) transgenic expresses the human diphtheria toxin receptor (hDTR) gene under the control of the HC-specific *myo6b* promoter. Treatment of adult heterozygous Tg(*myo6b*:hDTR) zebrafish with an injection of diphtheria toxin (DT) leads to widespread ablation of HCs in the saccule and utricle 5 days post injection without ablating neighboring *sox2* positive supporting cells[22] (Figure S1B). We characterized the gene expression profiles (scRNA-seq) and the map of accessible regions (scATAC-seq) associated with the response to HC ablation in adult zebrafish sensory epithelia. We dissected out saccules and utricles at three time points after ablation: days 4, 5, and 7 post-DT, and each sample was processed for either scRNA-seq or scATAC-seq experiments. Day 4 corresponded to maximal HC clearance after the apoptotic program had been initiated. Day 5 corresponded to when HCs remain absent, but regeneration has been clearly initiated. Day 7 corresponded to when HCs begin to repopulate sensory epithelia (Figures S1C and S1D). To control for the presence of the hDTR transgene and DT, untreated Tg(*myo6b*:hDTR) transgenic zebrafish, untreated wild-type fish, and wild-type DT-injected (day 4) zebrafish were used as non-regenerating controls. Subsequent single-cell analysis entailed pairwise comparisons between non-regenerating and regenerating sensory epithelia at each time point.

We generated 12 separate transcriptomic profiles on non-regenerating and regenerating inner ear tissues (saccule and utricle) using the 10x Chromium system for droplet-based scRNA-seq and quantified each dataset using the Cell Ranger 6.0.0 pipeline (10x Genomics). After filtering with Seurat,[23,24] we integrated scRNA-seq datasets from all 12 samples for cell type clustering to identify cell types based on conserved gene expression and to demonstrate that all 12 samples cluster together into a harmonized atlas of the adult zebrafish inner ear sensory epithelia.[23,24] Samples were assembled into an aggregate or "unified transcriptomic atlas" and unsupervised

clustering partitioned 66,296 inner ear sensory epithelial cells (saccule and utricle combined) into discrete scRNA-seq cell populations (Figure 1A; Table S6). We assigned cell type identities to clusters based on known expression of marker genes based on zebrafish transcriptome data from inner ear cells[25] and single-cell transcriptome data from the larval lateral line.[26,27]

For each cell population we identified genes specifically expressed or highly enriched (Figure 1B). Using Seurat pre-processing and integration procedures,[28] we selected genes that were variable between clusters. We assigned cell types closely related to HCs as: SCs and progenitor cells (PCs). We detected two populations of SCs: cluster 0 (SC0) and cluster 2 (SC2). Cluster 0 (SC0) showed significant enrichment for the markers *si:ch211-80h18.1* and *serpine2*, while cluster 2 (SC2) showed enrichment for the markers *krt18a.1* and *tnfrsf11b* (Figure 1D). We observed a distinct intermediate population of PCs which expressed *her4.1*, *her15.1*, and *dla* (Figure 1E). The PCs belong to clusters 1 (PC1) and 3 (PC3). This population of cells included both cycling (PC1) and non-cycling cells (PC3). Clusters 4 (HC4) and 5 (HC5) encompassed the HCs and expressed the *atoh1a*, *s100t*, and *pvalb8* genes (Figure 1F). No other cell cluster expressed HC lineage genes. Cluster 4 (HC4) was identified as young HCs because this cluster showed significant enrichment for the marker *rpl*, and GO analysis on genes showed enrichment for translation and ribosome assembly while cluster 5 (HC5) did not. Since synthesis of ribosomal proteins is shut down in mature HCs of the neuromast in the lateral line, cluster 5 (HC5) was identified as mature HCs.[25] Cells in all other clusters represented multiple non-sensory cell populations, such as immune cells, that do not directly contribute to HCs and these were excluded from downstream analyses regardless of whether they responded to HC injury.

### Adult inner ear cells are transcriptionally distinct from larval lateral line cells

The zebrafish lateral line has frequently been used as a model for HC regeneration, so we examined how much overlap the two organs showed in terms of the regenerative program. Adult inner ear sensory epithelia demonstrated distinct differences from previously fluorescence-activated cell sorting-purified larval lateral line neuromasts (Figure S2; Table S1).[27] There was strong separation of cell clusters and limited mixing of cells within clusters (Figures S2A and S2B). Figure S2C highlights the distinct gene expression profile of SCs, PCs, and HCs of adult inner ear. The results are consistent with significant transcriptional differences in cells between larval neuromasts and adult inner ears (Figure S2D).

### Similarities between gene expression in the saccule and utricle of the zebrafish inner ear

We compared saccule and utricle scRNA-seq experiments to identify cell types that are unique or shared between these two organs, to obtain conserved cell type markers and to also find cell-type-specific responses in regenerating sensory epithelia (Figure S3; Tables S2 and S3).[28] Despite different morphological profiles,[4,29] saccule and utricle sensory and non-sensory cells are transcriptionally similar and cluster tightly together (Figure S3A; Table S2). We tested if there were significant

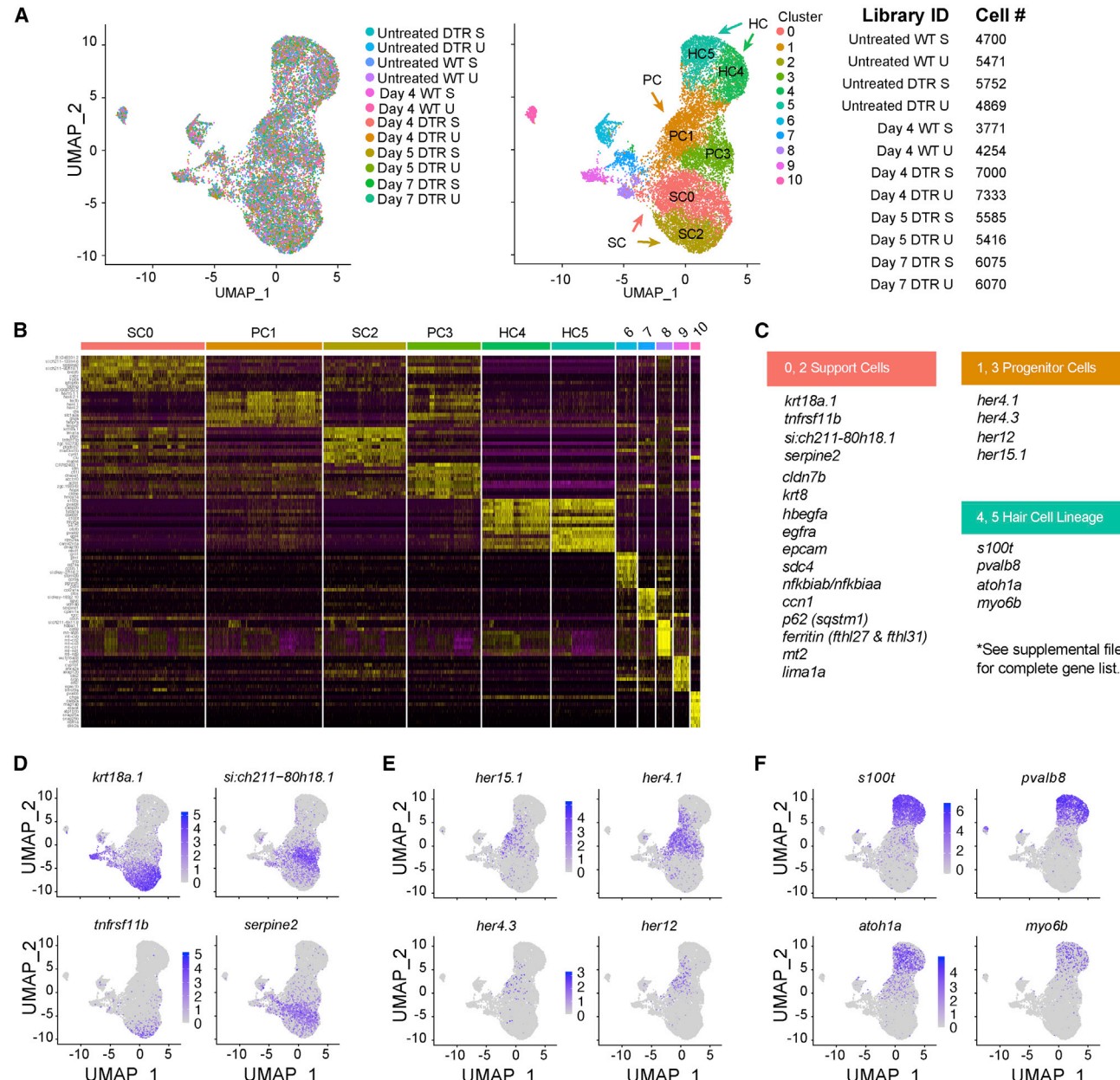

**Figure 1. scRNA-seq of the auditory and vestibular epithelia identifies gene expression**

(A) Left: scRNA-seq UMAP of 15,443 single cells collected from 12 samples. Each dot represents a single cell and colors distinguish samples. Middle: cells were clustered and labeled based on known cell markers. A single, joint clustering detects 11 cell populations. Clusters 0 to 5 consist of cells that contribute to the sensory epithelium: support cells (SCs) are clusters 0 (SC0) and 2 (SC2); progenitor cells (PCs) are clusters 1 (PC1) and 3 (PC3); and hair cells (HC) are clusters 4 (HC4) and 5 (HC5). Right: identifiers and cell counts for samples collected for scRNA-seq. S, saccule; U, utricle; DTR, heterozygous Tg(myo6b:hDTR) transgenic zebrafish; Day, number of days after DT injection, for a total of 66,293 sampled cells.

(B) Heatmap showing the relative expression levels of the top 10 differentially expressed genes (y axis) from each cluster (x axis). The cell population identity assigned to each cluster is indicated above each column and colors correspond to clusters in (B).

(C) Table of top significant marker genes.

(D–F) (D) UMAP plots of marker genes overrepresented in SCs, (E) PCs, and (F) HCs.

differences between regenerating saccule and utricle scRNA-seq experiments to determine if there was a divergence in the regeneration programs of the auditory and vestibular sensory epithelia (Figure S3B; Table S3). We looked broadly at changes

in gene expression between regenerating saccule and utricle cell types and found very few genes that differ (Figure S3C; Table S4). We found that gene expression changes are largely conserved between datasets for the three major cell populations

## A

Overlay

Merged

Non-Regenerating
Regenerating

Cluster
0 1 2 3 4 5 6 7 8 9 10 11 12

HC
HC2
PC1
PC
10
6 12
SC0
9 3
11 SC 7

## B

Non-Regenerating

Regenerating

Cluster
0 1 2 3 4 5 6 7 8 9 10 11 12

HC2
PC1
10
8
12
SC0
9 3
11 7

HC
HC2
PC1
PC
10
12
SC0
9 3
11 SC 7

Dynamics of Sensory Cell Populations

| Cell Type | Non-Regenerating | Regenerating |
|---|---|---|
| SC | 1303 | 576 |
| PC | 827 | 1119 |
| HC | 236 | 939 |

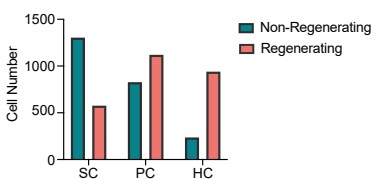

Non-Regenerating
Regenerating

## C

Regeneration Responsive Genes

HC
SC
1029
263
330
1388
484
265
407
PC

## D

six1b
pax2a
her6

Non-Regenerating
Regenerating

notch3
otogl
atoh1a

s100t
s100s
six1a

pou4f1
myo6b
sall1a

SC PC HC

*(legend on next page)*

that contribute to HC regeneration: SCs, PCs, and HCs (Figure S3C; Table S5).

We performed differential expression (DE) testing for cell types between saccule and utricle datasets using Seurat. In agreement with Yao et al.,[30] we observed elevated expression levels of *wnt11* (formerly *wnt11r*[31]), *sema3e*, *otol1a*, and *nr2f1*, and *vwa2* globally in or non-regenerating saccules but these genes are not differentially expressed in SC, PC, or HC clusters during regeneration (Table S5). Since the regeneration programs between organs were so similar, we combined saccule and utricle data for subsequent analysis to increase cell sample size and boost statistical power.

### Distinct regeneration responses of the zebrafish adult inner ear

Our scRNA-seq revealed regeneration-responsive transcription in a cell-type-specific manner. Using Seurat, we integrated non-regenerating controls and all regeneration conditions (days 4, 5, and 7) (Figure 2A, overlay). We then identified cell-type-specific responses to regeneration. We discerned 13 distinct cell populations (Figure 2A). Uniform Manifold Approximation and Projections (UMAPs) of individual cell types show that the cells in non-regenerating and regenerating tissues cluster together and all HC lineage populations were present both in homeostasis and during HC regeneration (Figure 2B). By taking an equal number of random cells from each sample, we assessed the number of cells in each cluster in regenerating sensory epithelia compared with non-regenerating controls. The regenerating inner ear exhibits a decline in the total number of SCs (SC0), an increase of PCs (PC1 in Figure 2B), and an increase of HCs (HC2) (Figure 2B). Despite clustering with the non-regenerating tissue, each cell type also possessed significant differences in the transcriptome compared with the homeostatic controls. All homeostatic control (untreated DTR fish and day 4 DT-injected wild-type fish) cell populations were essentially identical and ultimately pooled for further comparisons.

By performing DE testing for cell types between controls and regenerating datasets using Seurat, we identified 2,266 genes altered in expression between controls and regenerating SCs, 2,564 in PCs, and 3,164 in HCs (p < 0.1; FC ≥ 0.25). A total of 1,388 differentially expressed genes was shared among all three cell types (Figure 2C; Table S7). Global transcriptional changes in sensory epithelial cells involved in the HC lineage were observed following HC ablation (p < 0.01; FC > 0.25). The genes *pax2a* and *six1b* were upregulated in all regenerating sensory cell types: SC, PC, and HC. SCs and PCs showed an upregulation of *her6* (mammalian *Hes1*). PCs showed an upregulation of *notch3* and

*otogl1*. HC-specific genes, such as *atoh1a* and *s100t*, were upregulated in PCs and in HCs during regeneration. Finally, HCs were marked by an upregulation of *s100s*, *six1a*, *pou4f1*, and *myo6b* (Figure 2D; Table S7).

### Temporal gene regulation patterns in HC regeneration

By reversed graph embedding, Monocle 3 can measure cell fate changes in "pseudotime" (Figure 3A).[32] Monocle 3 grouped cells involved in HC regeneration and differentiation into 15 distinct subclusters and cell types were assigned to each cluster based on their gene expression profile (Figure 3B). Using the trajectory graph, we ordered the cells according to their progress through the regeneration program (Figure 3B). The SCs possessed the earliest developmental stage assignment and were designated as the root of the trajectory. Pseudotime was calculated for all other cell types based on their distance from the root of the trajectory (the SCs) and visualization of cells along the trajectory shows a transition between the two states (SC and HC). The populations serving as intermediate differentiation states connecting the SCs and HCs are the PCs.

Examples of genes that were expressed early (in SCs) and late (in mature HCs) in pseudotime in the UMAP cluster are shown in Figure 3C. We observed switch-like changes in expression of key regulatory factors, such as *cldn7b*, *her4.1* (*Hes5* in mammals), and *atoh1a*. The pseudotime ordering of cells showed that some genes act very early in SCs and then get shut off (like *cldn7b*), whereas others display dynamic temporal activity, turned on and then shut off in PCs and/or newly specified HCs.

We took all the genes that varied across the clusters and grouped those with similar patterns of expression into gene modules (Figure 3D). We identified nine co-expression modules, which represented genes that shared similar expression patterns during HC regeneration and showed genes that were upregulated at various stages of the regeneration process (Figure 3E; Table S8). The identified modules of co-regulated genes were specific to certain clusters of cells. We conducted GO enrichment analysis on each module (Table S8).

Module 1 was specific to immature HCs, while module 3 was specific to mature HCs. Genes grouped in module 1 were involved in HC differentiation and sensory perception of sound. Module 3 genes were involved in mRNA splicing, rRNA processing, and translation. Genes grouped in module 3 included many negatively regulated genes, such as *six1b* and *atoh1a*, which are known to be downregulated in mature HCs. Module 5 was specific to PCs and the genes were involved in ribosome biosynthesis. Module 2 was specific to SCs and PCs. Genes grouped in module 2 included both notch-independent (*hes2.2* and

---

**Figure 2. Cell-type-specific gene expression and expansion of HC lineage populations during regeneration**

(A) Left: UMAP showing overlay of inner ear cells between non-regenerating and regenerating sensory epithelia. Colors distinguish conditions. Right: UMAP of cells across conditions grouped together based on gene expression, detecting 13 cell populations. Cluster 0 consists of SCs (SC0). Cluster 1 consists of PCs (PC1). Cluster 2 consists of HCs (HC2).

(B) Left: UMAP plots of non-regenerating and regenerating conditions side-by-side. The colors distinguish clusters labeled in (A). Middle: UMAP of 3,670 randomly sampled single cells from non-regenerating controls (pooled wild-type, untreated Tg(*myo6b*:hDTR), and wild-type fish injected with DT). UMAP of 3,330 randomly sampled single cells from regenerating samples on days 4, 5, and 7 post injection. Samples include saccules and utricles. Right: table and bar graph of cell numbers in non-regenerating and regenerating SCs, PCs, and HCs.

(C) Venn diagram showing overlapping and unique differentially expressed genes in HCs, SCs, and PCs.

(D) Violin plots showing the distribution of gene expression of top genes identified with differential expression testing across cell types (p < 0.01; FC ≥ 0.25).

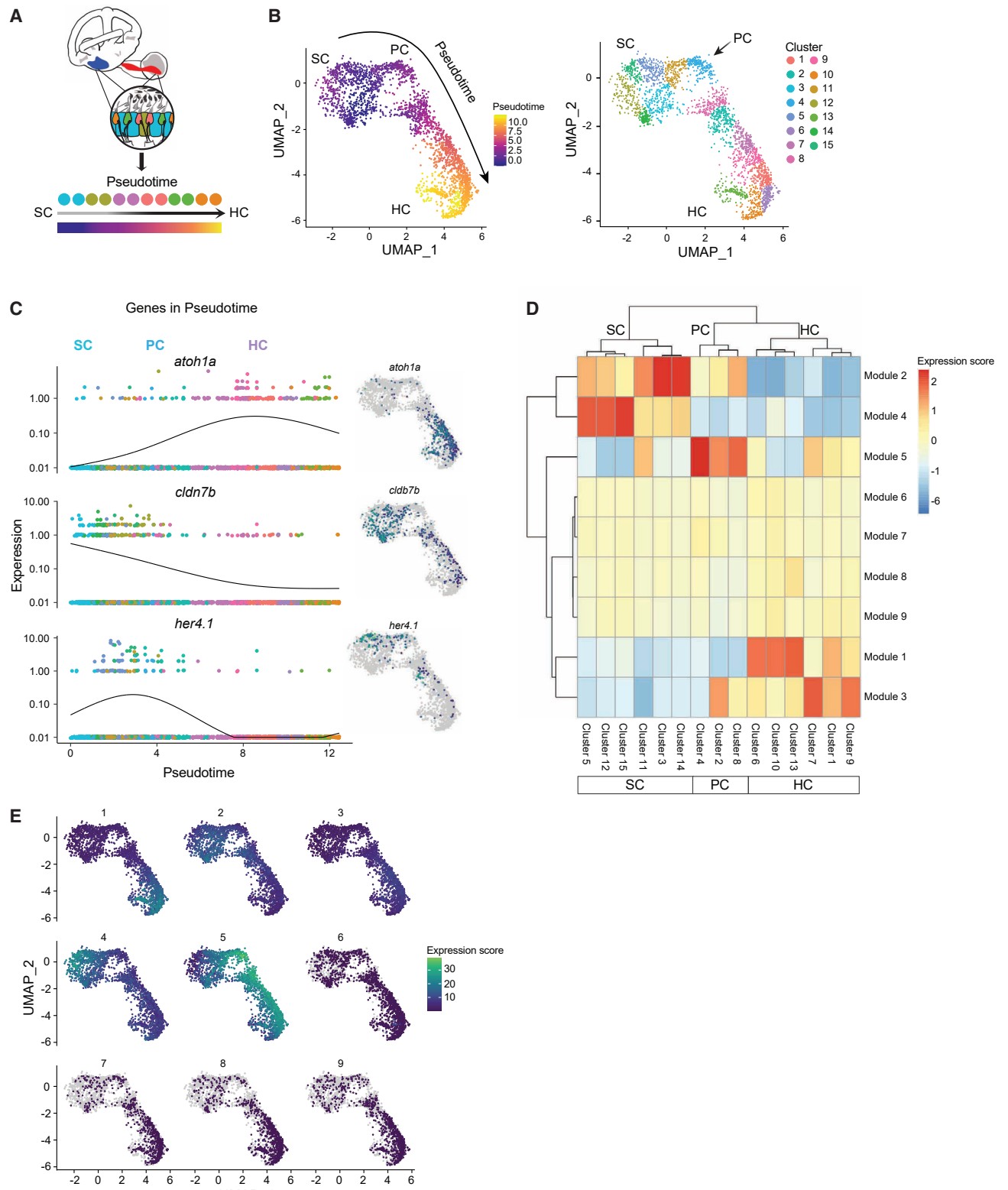

**Figure 3. Cell lineages in the inner ear**

(A) Schematic illustrating pseudotime cell ordering in the adult zebrafish inner ear. Inner ear HC lineage populations are colored according to their cluster membership and shown in pseudotime order.

*hes6*) and notch-dependent (*her4.1*, *her4.2*, *her4.3*, and *her4.4*) factors: *notch1a/b*, *sox10*, and *otogl.* The GO terms associated with module 2 were lateral line system development and cell junction organization/morphogenesis of epithelium. Module 4 was highly specific to SCs. Genes grouped in module 4 include mammalian *HES1*-related genes, *her6* and *her9*, with the associated GO terms negative regulation of transcription, implying the increase of *her* repressors in PCs during regeneration. We propose that modules 2 and 4 might be important for allowing the SCs and PCs to make cell-state transitions.[33]

### scATAC-seq reveals chromatin accessibility changes during regenerating

We performed scATAC-seq to explore chromatin accessibility of the regenerating adult zebrafish inner ear. We obtained 11 scATAC-seq profiles (Figure 4; Table S9). The quality of the dataset was assessed based on correlation with bulk ATAC-seq performed on similar samples. We found that the aggregate of scATAC-seq profiles closely resemble bulk ATAC-seq samples, indicating that the data were of sufficient quality.

We filtered and processed the resulting chromatin data.[24] We then integrated scATAC-seq datasets from all 11 separate samples for cell type clustering to identify cell types.[23,24] This resulted in a dataset with 11 clusters of single-cell epigenomes (Figure 4B, UMAP plot).

Using unsupervised clustering methods, nuclei were clustered and cell types were identified based on accessibility near expressed genes (Figure 4C; Table S10). Clusters were noisier than scRNA-seq measurements as scATAC-seq represents measurements from sparse chromatin data.[34] We integrated the scATAC-seq with scRNA-seq profiles using methods for cross-modality integration and label transfer (Figure 4D). scRNA-based classifications were consistent with the scATAC-seq UMAP visualization indicating shared correlation patterns and matched biological states across the two modalities (Figure 4D, right UMAP plot).

### Identification of regeneration-responsive elements

We identified regions that presented higher accessibility during regeneration compared with controls and named these regions regeneration-responsive elements (RREs). To characterize RREs, we obtained sc-ATAC-seq data from untreated inner ear saccules and utricles to represent the basal state of the tissues. We identified differentially accessible peaks between clusters. To identify RREs, we removed common peaks shared between untreated and treated samples leaving only the cell-type-specific emerging peaks.

Using this approach, we identified 12,369 RREs in SCs, 13,528 RREs in PCs, and 12,597 RREs in HCs (Figure 5A; Table S11).

The chromatin accessibility of PCs and HCs showed a much higher degree of overlap than the SCs did with either PCs or HCs. To understand the functions of emerging peaks, we applied GREAT analysis to annotate peaks and predict functions of putative regulatory regions (Figure S4).[35] GREAT analysis indicated that ∼19.5%–25.0% of peaks were located in proximal regions of the transcriptional start, whereas ∼70%–75% were located in distal regions (>±5 kb) (19.46% proximal PC peaks, 24.98% proximal SC peaks, 21.12% proximal HC peaks), many of which were in the first introns of genes. GO functional annotation of emerging peaks identified that peaks from all three cell types were associated with lateral line and inner ear development or differentiation, such as mechanosensory lateral line system development ($p \leq 6.2061e^{-6}$; SC peaks), mechanoreceptor differentiation ($p \leq 3.4662e^{-9}$; PC peaks), and inner ear receptor cell differentiation ($p \leq 1.4879e^{-8}$; HC peaks) (Figures S4D–S4F).

To determine whether cell-specific open chromatin regions from scATAC-seq analysis correlated with cell-specific gene expression, we developed a computational strategy to infer enhancer-to-gene relationships. We investigated to what extent enhancers in a window surrounding the transcriptional start site (TSS) of a gene (±50 kb from the TSS) predicted the expression of a gene (Figure 5B). Figure 5C shows an example for the *sox2* locus. For HCs 13,966 genes were associated with peaks, for SCs 13,709 genes were associated with peaks, and in PCs 13,874 genes were associated with peaks (Table S12).

We next correlated RREs with changes in gene expression during regeneration. We found a clear correlation between chromatin accessibility and nearby/distal gene expression where 60%–64% of differentially expressed genes were linked to RREs ($p < 0.05$, hypergeometric test) (Figures 5D and 5E). We identified, 1,546 differential genes (60.2% of DE genes) in PCs associated with at least one RRE peak ($p < 7.5 \times e^{-167}$, hypergeometric test),[36] 1,497 SC genes (60.0% of DE genes) ($p < 2.0 \times e^{-249}$, hypergeometric test), and 2,032 HC genes (64.2% of DE genes) ($p < 2.7 \times e^{-310}$, hypergeometric test) (Table S14). For further analysis, we chose the set of RREs that were linked to DE (regardless of whether that gene was the closest gene to the RRE) assuming that this subset would be enriched for regulatory regions that directly regulate gene expression.

### RREs are associated with conserved, non-coding elements

Conserved, non-coding elements (CNEs) are sequences outside of the coding regions that have a high degree of sequence conservation among multiple species. The "longer" a sequence is conserved across evolutionary time, the higher the probability it has a conserved functional purpose. Overlap of ATAC-seq

(B) Left: cells progressing through the HC regeneration program. The arrow indicates the trajectory of the pseudotime differentiation gradient from purple to yellow. SCs are the root node of the trajectory graph, PCs emerge after HC injury and are a clear intermediate to both SCs and HCs. Right: UMAP showing the separation of HC lineage populations into groups after re-clustering for pseudotime (resolution = $1 \times e^{-2}$ for Louvain clustering). Subclusters 3, 5, 11, 12, 14, and 15 belong to SCs. Subclusters 2, 4, and 8 belong to PCs. Subclusters 1, 6, 7, 9, 10, and 13 belong to HCs. The arrow indicates the first PC cluster (cluster 4) that is transcriptionally distinct from SCs.

(C) Gene expression dynamics of select genes as a function of pseudotime for SCs, PCs, and HCs. Colors distinguish subclusters belonging to SC, PC, and HC populations. Adjacent UMAP plots of select genes that are differentially expressed through the trajectory.

(D) Clustered heatmap of gene modules co-regulated along the pseudotime during HC initiation, progression, and maturation. Nine modules and their expression intensity in each cluster are shown. Clusters correspond to cell populations as indicated on the horizontal axis.

(E) Maps showing modules expressed in specific clusters, while other modules are shared across multiple states of pseudotime.

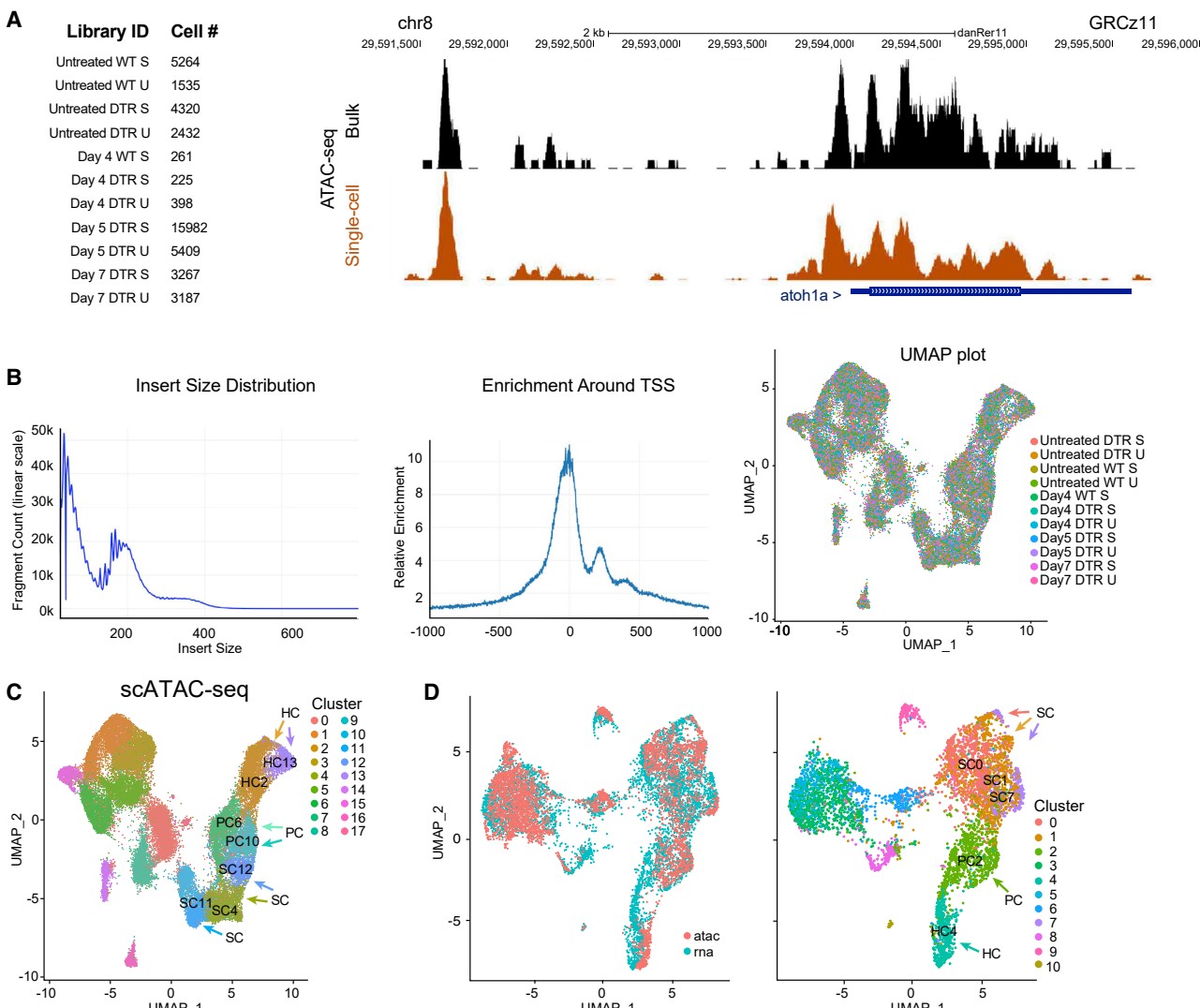

**Figure 4. scATAC-seq of auditory and vestibular epithelia during regeneration**

(A) Left: identifiers and cell counts for samples collected for scATAC-seq. S, saccule; U, utricle; DTR, heterozygous Tg(*myo6b*:hDTR) transgenic zebrafish; and Day, number of days after DT injection (total = 42,278 cells). Right: genome browser track of the *atoh1a* locus highlighting annotated peaks comparing single-cell data with bulk ATAC-seq.

(B) Left: the library fragment size distribution displayed nucleosome banding patterns. Middle: sequencing reads showed strong enrichment around transcriptional start sites (TSS). Right: high sample agreement is shown between all samples.

(C) Aggregate of all scATAC-seq samples, including untreated samples. Clusters 4, 11, and 12 consist of SC: SC4, SC11, and SC12. Clusters 6 and 10 consist of PCs: PC6 and PC10. Clusters 2 and 13 consist of HCs: HC2 and HC13.

(D) Left: UMAP co-embedding shows scATAC-seq and scRNA-seq cells on the same plot. Right: RNA cluster groups maintain cohesion in ATAC-seq data. SC populations are clusters 0, 1, and 7: SC0, SC1, and SC7. PC cell population is cluster 2: PC2. HC population is cluster 4: HC4.

peaks with CNEs implies functional significance.[37] To further validate these data and the likelihood of identified enhancers for functional roles *in vivo* and to also make use of evolutionary sequence conservation as a filter to find putative gene regulatory elements, we intersected identified RREs associated with differential gene expression with CNE data derived from comparisons between zebrafish and several carp species generated by Chen et al.[38] (Figure 5D; Table S13). We found significant overlap between conserved non-coding blocks and RRE peaks associated with at least one differentially expressed gene identified in our

scATAC-seq and scRNA-seq data (SC: $p < 2.9 \times e^{-178}$; PC: $p < 3.5 \times e^{-134}$; HC: $p < 2.0 \times e^{-193}$, hypergeometric test) (Figure 5E; Table S14). Together these data identified *cis*-regulatory elements strongly enriched for roles in transcriptional control during HC regeneration.

## Deep learning identifies key regulatory motifs and potential co-regulation cassettes

Machine learning can accurately identify enhancer sequences *de novo* from raw sequence data.[39] Therefore, we constructed

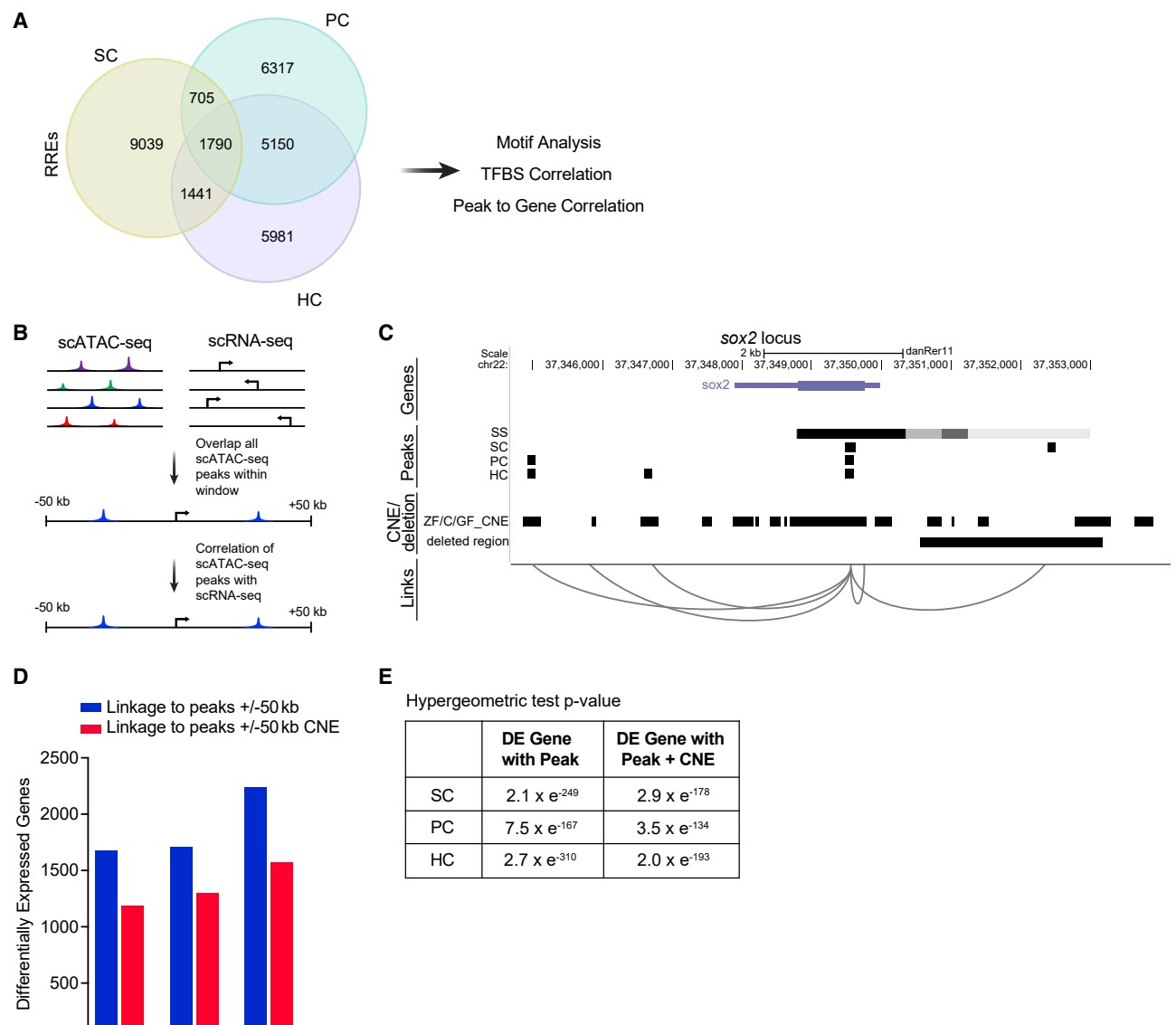

**Figure 5. Correlation of scATAC-seq peaks to conserved regions and to differentially expressed genes**
(A) Venn diagram of regeneration-responsive elements (RREs) identified in SCs, PCs, and HCs.
(B) Schematic of the approach used to link scATAC-seq peaks in proximal and distal DNA elements to genes.
(C) Schematic of predicted links between peaks near the gene, *sox2*. For each cell type, a track is shown, including peaks present in non-regenerating controls (SS, steady state), conserved non-coding elements (CNEs) from genomic four-way phastcons (Z, zebrafish; C, grass and common carp; GF, goldfish), and the deleted region in *sox2*^hg138^ mutants. Genome coordinates of peaks, CNEs, and deleted region are represented by black bars.
(D) Bar graphs showing the number of RREs that are within 50 kb of differentially expressed genes (blue) and overlap with CNEs (red) in each cell type.
(E) Intersections of differentially expressed (DE) genes with peaks and peaks with CNEs are statistically significant (p < 0.001).

a prediction model to determine if enriched transcription factor binding sites (TFBSs) were found in the RREs. A deep learning (DL) model using a 59,785-parameter, 4-layer DL model trained by integrating features associated with enhancer activity from Danio-CODE,[40] and these models were then applied to RREs (Table S15). The model showed that Six and Sox motifs were significantly enriched in our datasets in a cell-specific manner (Figure 6A). Furthermore, DL revealed that Sox factors were

more strongly enriched in SCs (Figure 6B) and showed significant co-occurrence with Pdx1, Foxd3, Cebpa, Ebf1, and Rest motifs. In PCs, Sox and Six sites were significantly enriched and both often co-occurred with Creb1 or Fos/Ap1 but not with each other. In HCs, Sox sites showed significant co-occupancy with Six4 as well as with Creb1, and Fos/Ap1 (correlation of TFBSs p < 0.001) (Figure 6A). We parsed the RREs into four categories, containing only Sox motifs, containing only Six

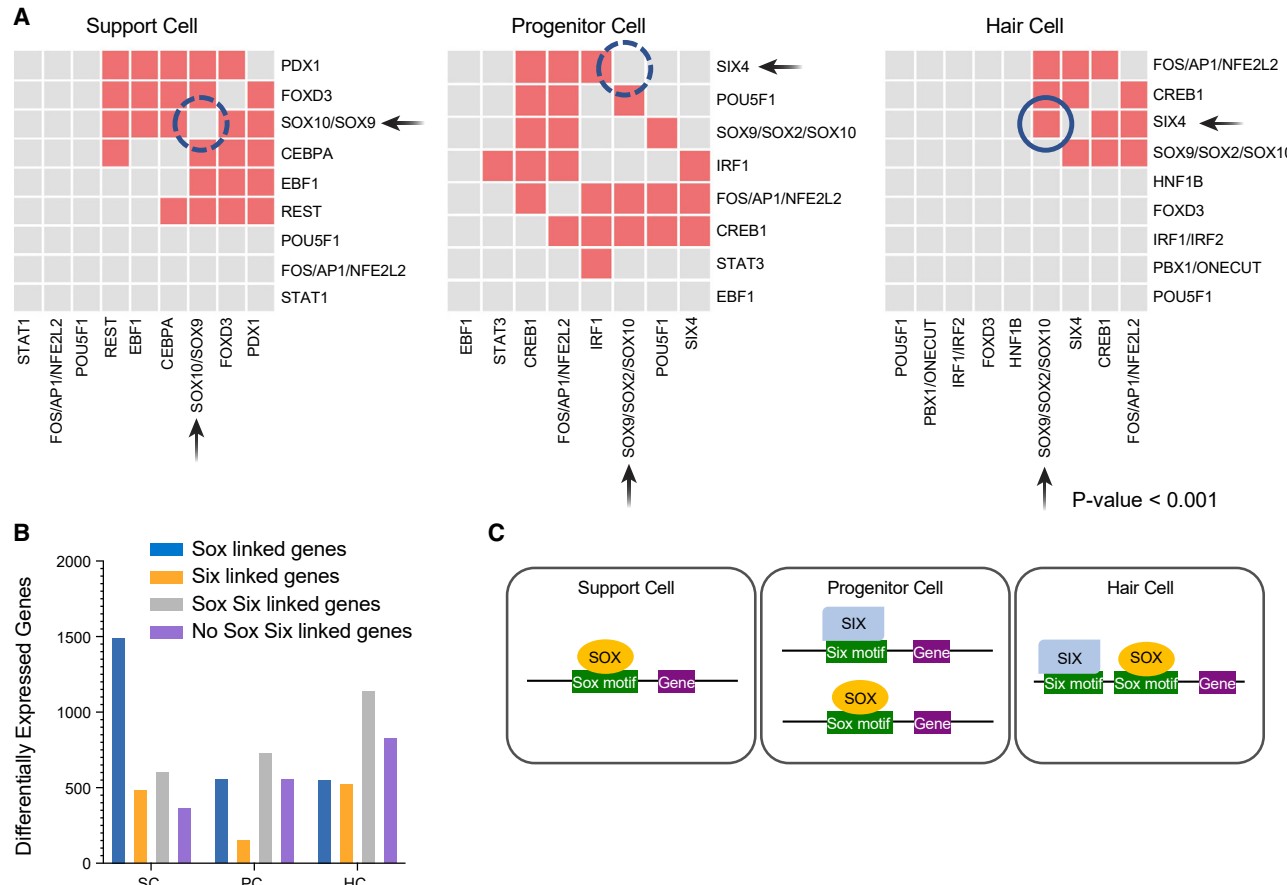

**Figure 6. Combinatorial accessibility of Six and Sox TFs are defined by distinct cell identities**

(A) Pairwise predicted TF co-association in SCs, PCs, and HCs. x and y axes show significantly enriched predicted TF binding sites. Gray boxes represent lack of co-association between the TFs, whereas the colored boxes represent co-association. Correlation of TFBSs p < 0.001.

(B) Sox and/or Six TF motif containing enhancers associated with differentially expressed genes in SCs, PCs, or HCs. Categories of TF containing motifs include Sox containing, Six containing, Sox and Six containing, or neither.

(C) Model of Six and Sox TFs with cooperative roles functioning as either stem cell promoting or HC promoting factors.

motifs, containing both Sox and Six motifs, or containing neither Sox nor Six motifs (Figure 6B; Table S16). We detected a clear pattern in SCs, where 71% of all enhancers linked to differentially expressed genes (1,498 genes) in the SCs had Sox motifs. PCs, despite having more total peaks than SCs, had 944 fewer gene associations total with no clear pattern of preference across the four categories. This pattern may be consistent with a cell type that is in transition. HCs had the most total differentially expressed genes associated with enhancer elements, with the largest category being peaks containing both Sox and Six motifs (37%). The second largest category for HCs had neither Sox nor Six motifs. In SCs, the top GO enrichment terms for Sox-linked genes were: "negative regulation of protein processing," "negative regulation of protein maturation," "hemidesmosome assembly," and "regulation of epidermis development" consistent with epithelial cells that would be undergoing a state transition. Highest GO enrichment for the HC Six/Sox-linked genes were: "peripheral nervous system axonogenesis," "peripheral nervous system development," and "peripheral nervous system differentiation" consistent with driving HC fates.

Based on the scRNA-seq data and DE testing, we determined which Sox and Six factors were expressed in each cell type to correlate accessible binding sites to available transcription factors (TFs) (Figure S5A). In agreement with the pseudotime analysis, the Sox genes were dynamically expressed (Figure S6). The Sox genes *sox4a*, *sox4b*, *sox11a*, and *sox21a* all showed DE changes, as did *six1a*, *six1b*, *six4a*, and *six4b* (p < 0.01; FC ≥ 0.25) (Figure S5B). Genes, such as *sox10* and *sox11a*, are SC markers. In SCs, although *sox10* was virtually unaffected, *sox11a* (p < 1.64 × e$^{-5}$; FC = −0.77) and *sox4b* (p < 1.04 × e$^{-20}$; FC = −1.48) had reductions in expression during HC regeneration. *sox4b* continued to be reduced in expression in PCs (p < 2.03 × e$^{-12}$; FC = −0.79) and HCs (p < 1.04 × e$^{-20}$; FC = −1.48). The changes in *sox2* and *sox21a* were subtle and particularly restricted to the PCs according to pseudotime analysis (Figure S6), suggesting that they are key drivers of the state change, while both *six1* genes, but particularly *six1b* in both regenerating and non-regenerating sets (p < 2.34 × e$^{-6}$; FC = 0.57), were very strongly expressed in HCs (p < 0.01; FC > 0.25) (Figure S5B).

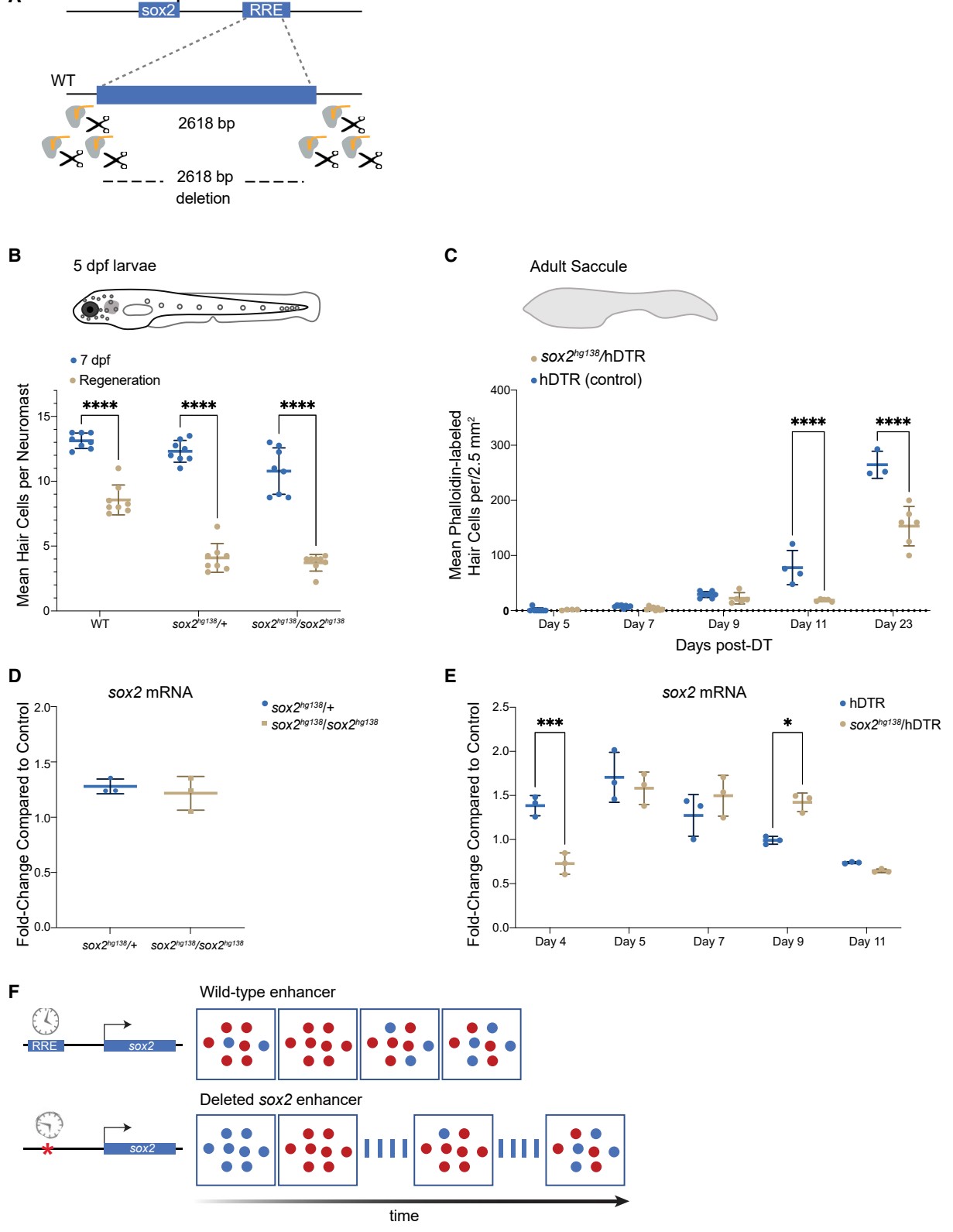

*(legend on next page)*

Six family members have a demonstrated role in mammalian HC differentiation[41] and a knockout of *Six1* results in defective inner ear development.[42] In mice, *Sox2* has known roles in sensorineural development and is expressed in HC PCs and SCs until it is downregulated after differentiation.[43] The observed pattern of motif enrichment suggests a significant shift from Sox-driven gene expression in SCs to Sox/Six co-regulated expression in HC. The relatively smaller number of differentially expressed genes linked to the emerging peaks in the PCs is consistent with them being a transient state between the SCs and HCs, where the chromatin is opening to allow for gene expression, but actual gene activation has not occurred yet. Once gene expression is initiated, the cells shift to an HC classification.

### Deletion of a regulatory element of sox2 results in a dominant loss of HC regeneration

Our integrated single-cell analysis predicted that *sox2* is differentially regulated in a switch-like pattern such that *sox2* expression increases in PCs and then shuts off in newly differentiating HCs (Figure S6). Based on the known role of *sox2* in stem cell pluripotency in general and its role in inner ear development specifically, we were interested in identifying regulatory elements for *sox2* that were specifically activated during hearing regeneration and if *sox2* was a key driver of regeneration. We found the *sox2* locus acquired cell-specific dynamic changes in accessibility during HC regeneration with ATAC accessibility of specific regions correlating to upregulation in gene expression (Figure 5C). A specific element 1,995 bp upstream the TSS of *sox2* emerged in SCs and PCs only during regeneration (but was absent in the HCs), suggesting that this might represent an enhancer of *sox2* with important roles in regulating regeneration (Figure S7).

A 2,115 bp peak region overlapped with a CNE identified from zebrafish/goldfish/carp alignments[38] as well as three shorter "ultraconserved" regions present in all vertebrates[44](Figure S7). In addition, the JASPAR database[45] predicted multiple TFBSs, including sites for *sox2*, *pou4f3*, and *stat3*, all of which are associated with inner ear development and/or regeneration.

To functionally validate if this region regulates *sox2* during regeneration, we utilized CRISPR-Cas9 to introduce deletions of the 2 kb *sox2* upstream enhancer. We injected spCas9 protein with six guides flanking the region into single-cell-stage embryos (Figure 7A). We identified germline transmitted deletions and generated three independent enhancer deletion lines (Figure S9A). The majority of the experiments were performed on one stable line of heterozygous enhancer deletions or homozygous deletions, denoted here as *sox2^hg138*. Results were verified in the two other alleles denoted as *sox2^hg139* and *sox2^hg140* (Figure S9A).

Enhancer deletion mutants had no overt morphological phenotypes in early larvae, and both heterozygous and homozygous deletions survived to adulthood. Homozygous and heterozygous fish were viable as adults and the overall morphologies of inner ears, sensory epithelia, lateral line neuromasts, and even swimming behaviors appeared normal for all three enhancer deletion alleles.

We performed larval lateral line regeneration assays on the enhancer deletion mutants using $CuSO_4$ ablation.[46] We first quantified the mean HCs per neuromast in untreated heterozygous and homozygous enhancer deletion mutants 7 days post fertilization and found no differences from wild-type for both *sox2^hg139* and *sox2^hg140*, either as heterozygous or homozygous deletions, and no difference for *sox2^hg138* heterozygous deletions. The *sox2^hg138* is the largest deletion (Figure S9) and we did find a statistically significant difference between wild-type and homozygous *sox2^hg138* enhancer deletion mutants with the mutants having on average one to two fewer HCs per neuromast (two-way ANOVA, $p < 0.001$) (Figure S9).

We found that both homozygous and heterozygous deletions had no effect on larval HC development (or modest effects in *sox2^hg138*/*sox2^hg138* homozygous mutants), but all deletions altered HC regeneration with similar severity regardless of whether the deletions were in one copy or two (Figures 7B and S9). Therefore, the enhancer region was a haploinsufficient regulator of neuromast HC regeneration.

---

**Figure 7. The −1,995 bp *sox2* enhancer element is required for HC regeneration**

(A) Generation of enhancer deletion mutants using CRISPR-Cas9 editing.

(B) Lateral line HC regeneration is strongly inhibited 2 days after $CuSO_4$ ablation in homozygous (*sox^hg138*/*sox^hg138*) and heterozygous (*sox^hg138*/+) enhancer deletion mutants. The average number of HCs and SD are shown in the graph. A two-way ANOVA comparison and Sidak multiple comparison test of the data obtained on untreated larvae with the data obtained from $CuSO_4$ treated larvae ****$p < 0.0001$. Error bars show SD. n = 8 larvae in each group.

(C) Adult HC regeneration is significantly inhibited up to 23 days after HC ablation in heterozygous (*sox^hg138*/hDTR) enhancer deletion mutants. The average number of HCs is shown in the graph. A two-way ANOVA comparison of the data obtained on regenerating control Tg(*myo6b*:hDTR) saccule (hDTR, blue) with the data obtained from regenerating heterozygous enhancer deletion mutant (*sox^hg138*/hDTR, beige) saccule and Sidak multiple comparison test: ****$p < 0001$. Error bars show SD. n = 6–8 saccules in each group unless otherwise indicated.

(D) Quantitative real-time PCR measuring *sox2* mRNA levels in adult homeostatic sensory epithelia (saccule) from heterozygous (*sox2^hg138*/+, blue) and homozygous (*sox2^hg138*/*sox2^hg138*, beige) enhancer deletion mutants. Fold-change compared with homeostatic sensory epithelia (saccule) of wild-type controls of the same age.

(E) Quantitative real-time PCR measuring *sox2* mRNA levels in regenerating sensory epithelia (saccule) from Tg(*myo6b*:hDTR) controls (hDTR, blue) and sensory epithelia from heterozygous enhancer deletion mutants (*sox2^hg138*/hDTR, beige) shows that activation of *sox2* expression is delayed by 24 h (day 4) then remains elevated for an additional 24 h (day 9). The delay in *sox2* expression at day 4 and the persistent expression at day 9 are statistically significant. A two-way ANOVA comparison of the data obtained on regenerating control (hDTR, blue) saccule with the data obtained from regenerating heterozygous enhancer deletion mutant saccule (*sox^hg138*/hDTR, beige) and Sidak multiple comparison test: *$p < 0.02$, ***$p < 0.0004$. Error bars show SD and triplicate technical replicates from dissected saccule of six to eight adult fish are shown in the graph.

(F) The upstream enhancer is involved in regulating the timing of *sox2* expression but not essential for triggering activation. *sox2* gene activation occurring in the supporting cells is depicted by the appearance of red dots. *sox2* levels are delayed in enhancer deletion mutants compared with wild-type, but reaches the appropriate levels 24 h later. The return to baseline expression is also delayed by 24 h.

To examine the role for the *sox2* upstream enhancer deletion in adult zebrafish, we again employed the Tg(*myo6b*:hDTR) transgenic zebrafish[22] (Figure 7C). In our experiments, heterozygous *sox2*[hg138]/Tg(*myo6b*:hDTR) and Tg(*myo6b*:hDTR) zebrafish were administered DT, and sensory epithelia were examined after 7, 9, 11, and 23 days of recovery following injection. In untreated heterozygous *sox2*[hg138]/Tg(*myo6b*:hDTR), saccular HCs were present in normal numbers (Figure S8). In DT-injected heterozygous *sox2*[hg138]/Tg(*myo6b*:hDTR) zebrafish, there was a major reduction in HC regeneration after DT treatment on day 11 post DT in comparison with normally regenerating sensory epithelia (two-way ANOVA, $p < 0.0011$). Adult HC regeneration continued to be significantly inhibited 23 days after HC ablation in heterozygous *sox2*[hg138]/Tg(*myo6b*:hDTR) zebrafish (two-way ANOVA, $p < 0.0001$). Our data reveal that the −1,995 bp upstream enhancer of *sox2* is required specifically for HC regeneration but not normal HC development in zebrafish and heterozygous deletions possessing a phenotype as severe as homozygous deletions. We did not notice any regeneration deficits after tailfin amputation in adult heterozygous *sox2*[hg138]/Tg(*myo6b*:hDTR) fish, suggesting that the regeneration phenotype for this deletion was restricted to HC regeneration, or perhaps to neuronal regeneration in general.

### The −1,955 bp sox2 enhancer regulates the timing of sox2 expression during regeneration

We sought to determine if *sox2* expression levels were altered in heterozygous *sox2* enhancer deletion fish compared with Tg(*myo6b*:hDTR) controls during regeneration. *sox2* mRNA levels were measured in sensory epithelia by quantitative real-time PCR analysis. In non-regenerating sensory epithelia of the heterozygous enhancer deletion mutants (*sox2*[hg138]/+), *sox2* RNA levels were not significantly altered in comparison with control wild-type sensory epithelia (Figure 7D). In adult Tg(*myo6b*:hDTR) zebrafish without the enhancer deletion undergoing HC regeneration, *sox2* expression was elevated on days 4, 5, and 7 post DT. This was consistent with our single-cell transcriptomics assays on regenerating inner ear tissues collected at all time points (days 4, 5, and 7) (Figure S5). By day 9 post DT, *sox2* expression levels were near control levels and were further reduced on day 11 post DT (Figure 7E).

In heterozygous enhancer deletion mutants, we found that *sox2* expression levels were significantly reduced on day 4 in comparison with normal inner ear sensory epithelia undergoing regeneration (two-way ANOVA, $p < 0.0039$). By day 5–7 post-DT, *sox2* expression levels rose to levels comparable with normally regenerating inner ears. At day 9 post DT, levels of *sox2* remain elevated in enhancer deletion mutants (two-way ANOVA, $p < 0.0021$) in comparison with wild-type, but by day 11 post DT, levels of *sox2* did drop back to levels comparable with wild-type. Our data suggest that the upstream enhancer of *sox2* may be specifically involved in regulating the timing of *sox2* expression but not essential for triggering activation (Figure 7F). It is interesting to note that, despite *sox2* levels apparently only being shifted by 24 h in the mutant compared with the wild-type, the regeneration of HCs appeared to have been affected out to at least 23 days post ablation and potentially permanently.

## DISCUSSION

Regeneration of inducible or injury-dependent gene expression may be controlled by specific enhancer regulatory elements.[7–9,12–15,17,18,21] Tissue regeneration enhancer elements (TREEs) have been identified by approaches, such as H3.3 profiling and epigenetic profiling. While validation of identified regulatory elements revealed that the putative enhancers can direct expression of minimal promoters or reporters, deletion of multiple putative enhancers from the genome previously resulted in no detectable effects on regeneration.[7,13]

### Origin of HCs during zebrafish inner ear HC regeneration

Our regeneration assay causes selective HC ablation, leaving the population of Sox2-positive supporting cells undamaged (Figure S1). Similar to what is observed in the regenerating chick cochlea following acoustic trauma[47] and consistent with clonal analysis of HC origins in the chicken hearing organ the basilar papilla and lateral line organs in zebrafish,[48,49] our single-cell trajectory analysis predicts that supporting cells contribute new HCs in the zebrafish inner ear by reverting to a less-differentiated state we have labeled as PCs (Figure 3B). The PCs are transcriptionally distinct from SCs and HCs and represent a special transition state cell type that may reenter mitosis when HCs are depleted and become a new HC. In addition, we find that regenerating sensory epithelia exhibit a decline in SCs and an increase in PC and HC number (Figure 2B). From this analysis it is unclear if the PCs emerge from the SCs and can return to SC identity or if there are resident PCs that rapidly expand after an injury and either become HCs or SCs. Even in uninjured inner ears, we detected PCs. We can envision two reasons why this may be (1) it has been shown in adult zebrafish that HCs continually grow throughout the lifespan,[50] the PCs could be always present because they are in the process of transitioning from SCs to HCs during that growth, or (2) the PCs in fish represent a cache of multipotent stem cells that contribute to the constant growth and can also replenish SCs and/or HCs after injury or death. Understanding how SCs give rise to new HCs in the adult inner ear of zebrafish may contribute to a better understanding of overcoming the regeneration block in the mammalian inner ear.

There is some evidence for the presence of inner ear cell types with the capacity for self-renewal and HC-like formation in the mammalian vestibular organ of mice.[51] However, the adult cochlea completely lacks the capacity to regenerate due to the loss of a stem cell population as the mammal ages.[52] The continued effort to reprogram differentiated mammalian inner ear cells may require comparing how SCs propagate new HCs in response to damage or how to drive SC identity back to the progenitor state seen in regenerating model systems. Identifying enhancers that are key regulators of regeneration are an essential step in identifying these differences in regenerative capacity.

### Six and Sox TFs appear to act in a combinatorial fashion to drive cellular identity

The *Sox* and *Six* genes are known to be among highly expressed TFs important for HC development and differentiation.[53] Our analysis revealed that HC regeneration appears to be orchestrated by specific combinations of Six and Sox. These TFs may

**Cell Genomics**
Article

have cooperative roles in mediating activity of individual regulatory elements and may function as either stem cell-promoting or HC-promoting factors (Figure 6C) depending on context. In mice, prosensory cells (Sox2-EGFP+) from the embryonic cochlear duct are enriched for motifs corresponding to Six, Sox, Gata, Ebf, and Tead families, as well as motifs for Grhl2, Lef1, Irf4, and Rest.[54] We identified 30,423 open chromatin regions enriched for many of the same TFs. Motif enrichment and co-occupancy in our data provides evidence for shifting roles of Six and Sox at different stages of HC regeneration and implicates several other TF families during regeneration.

There are multiple Sox and Six factors expressed during regeneration suggesting some possible redundancy of function. In the case of Sox2 function, it was notable that simply delaying expression by 24 h is sufficient to severely inhibit regeneration, suggesting that, if there is some redundancy with other Sox TFs, it is not nearly sufficient to compensate for the missing Sox2 function. It also suggests merely ectopically activating genes in the inner ears of mammals may be insufficient to induce a proper regeneration response.

### Enhancer regulatory elements with roles in regeneration

Regeneration-competent animals are likely to possess conserved genomic regions necessary to activate injury and regeneration responses[21]; we found there was a large degree of overlap between conserved non-coding sequences between zebrafish and carp with RRE peaks associated with differentially expressed genes. We identified thousands of RREs that have been conserved for more than 60 million years of divergent evolution.

Previously generated enhancer deletions in zebrafish and *Drosophila* have had no detectable effects on regeneration-responsive gene expression nor resulted in a regeneration defect.[7,17] In this study, we characterized regions that emerge in response to HC regeneration and identified an upstream enhancer of *sox2*. A surprising finding is that deletion of this region in either single or double knockouts resulted in the same level of abnormal regeneration and misregulated expression of *sox2*. This deletion demonstrated that the upstream enhancer was genetically required for normal HC regeneration and controlled the timing of *sox2* expression. These observations were true for both the inner ear sensory epithelium and the lateral line. Our results demonstrate that not only are changes in *sox2* expression an essential component of hearing regeneration, initiation of *sox2* expression alone was insufficient for regeneration and that the timing of initiation and the duration of expression for the *sox2* TF were also critical factors for successful hearing regeneration. Addressing if inner ear regeneration enhancer elements are different among species with different regenerative capacities and that the capacity to regenerate a given tissue might be impacted by these sequence differences will provide insight on how to proceed in "reactivating" the regeneration response in mammals.

### Limitations of the study

A potential limitation of the study is that the scATAC-seq and scRNA-seq datasets originated from two different experi-ments, so correlations between cell clusters must be implied by linking promoter sequences from the scATAC-seq data to changes in gene expression in the scRNA-seq data. Also the current data cannot distinguish between two models of regeneration: (1) SCs differentiating into PCs then HCs, or (2) resident PCs being able to differentiate into either HCs or SCs.

### STAR★METHODS

Detailed methods are provided in the online version of this paper and include the following:

- KEY RESOURCES TABLE
- RESOURCE AVAILABILITY
  - Lead contact
  - Materials availability
  - Data and code availability
- EXPERIMENTAL MODEL AND SUBJECT DETAILS
- METHOD DETAILS
  - Adult zebrafish diphtheria toxin administration
  - Fine dissection of adult inner ear for single cell suspensions
  - Single-cell suspensions on adult sensory epithelia
  - 10x genomics scRNA-seq library construction
  - 10x genomics scATAC-seq library construction
  - CRISPR/Cas9 enhancer deletion
  - RNA isolation from adult inner ear tissues
  - Quantitative real-time PCR analysis on adult inner ear tissues
  - Lateral line hair cell and neuromast quantification
  - Histological methods for imaging
  - Adult inner ear hair cell labelling
  - Cellular imaging and analysis of inner ear hair cells
- QUANTIFICATION AND STATISTICAL ANALYSIS
  - scRNA-seq data processing
  - scRNA-seq analysis with Seurat
  - Cell numbers in cell populations of homeostatic and regenerating sensory epithelia
  - Pairwise comparisons and DE testing using Seurat
  - Pseudotime analysis using Monocle 3
  - Finding modules of co-regulated genes using Monocle 3
  - scATAC-seq data processing
  - scATAC-seq analysis with Signac and Seurat
  - scRNA-seq and scATAC-seq integration
  - Identification of RREs
  - GREAT analysis and gene ontology annotations
  - Motif analysis using HOMER
  - Peak-to-gene annotation
  - Hypergeometric test
  - Identification of co-occurrence of *de novo* motifs using deep learning (DL)

### SUPPLEMENTAL INFORMATION

## ACKNOWLEDGMENTS

This research was supported by the Intramural Research Program of the National Human Genome Research Institute (ZIAHG200386-06). We thank Dr. Tannia Clark and Charles River for zebrafish care; Suiyuan Zhang and the Bioinformatics and Scientific Programming Core; Stephen Wincovitch and the NHGRI Cytogenetic and Microscopy Core Facility; Martha Kirby, Stacie Anderson, and NHGRI Flow Cytometry Core Facility; Di Huang for her valuable suggestions on computer programming and the computational resources of the NIH HPC Biowulf cluster (http://hpc.nih.gov); Julia Fekecs, Darryl Leja, and the Office of Communications at NHGRI; and the members of the Burgess laboratory for helpful discussion. All animal experiments were approved by the National Human Genome Research Institute's Animal Care and Use Committee (protocol #G-01-3).

## AUTHOR CONTRIBUTIONS

Conceptualization, E.J. and S.M.B.; methodology, E.J. and S.M.B.; investigation, E.J., C.C.S., W.S., W.W., A.G.E., Z.C., D.G., I.O., and S.C.F.; visualization, E.J.; funding acquisition, S.M.B. and I.O.; writing – original draft, E.J. and S.M.B.; writing – review & editing, E.J., C.C.S., W.S., Z.C., S.C.F., D.G., W.W., A.G.E., I.O., and S.M.B.

## DECLARATION OF INTERESTS

The authors declare no competing interests.

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

## Article

58. Varshney, G.K., Carrington, B., Pei, W., Bishop, K., Chen, Z., Fan, C., Xu, L., Jones, M., LaFave, M.C., Ledin, J., et al. (2016). A high-throughput functional genomics workflow based on CRISPR/Cas9-mediated targeted mutagenesis in zebrafish. Nat. Protoc. *11*, 2357–2375. https://doi.org/10.1038/nprot.2016.141.

59. Goudarzi, M., Berg, K., Pieper, L.M., and Schier, A.F. (2019). Individual long non-coding RNAs have no overt functions in zebrafish embryogenesis, viability and fertility. Elife *8*, e40815. https://doi.org/10.7554/eLife.40815.

60. Brownstein, M.J., Carpten, J.D., and Smith, J.R. (1996). Modulation of non-templated nucleotide addition by Taq DNA polymerase: primer modifications that facilitate genotyping. Biotechniques *20*, 1008–1010. https://doi.org/10.2144/96206st01.

61. Livak, K.J., and Schmittgen, T.D. (2001). Analysis of relative gene expression data using real-time quantitative PCR and the 2(-Delta Delta C(T)) Method. Methods *25*, 402–408. https://doi.org/10.1006/meth.2001.1262.

62. Pei, W., Xu, L., Varshney, G.K., Carrington, B., Bishop, K., Jones, M., Huang, S.C., Idol, J., Pretorius, P.R., Beirl, A., et al. (2016). Additive reductions in zebrafish PRPS1 activity result in a spectrum of deficiencies modeling several human PRPS1-associated diseases. Sci. Rep. *6*, 29946. https://doi.org/10.1038/srep29946.

63. Liang, J., and Burgess, S.M. (2009). Gross and fine dissection of inner ear sensory epithelia in adult zebrafish (Danio rerio). J. Vis. Exp. https://doi.org/10.3791/1211.

64. Liang, J., Wang, D., Renaud, G., Wolfsberg, T.G., Wilson, A.F., and Burgess, S.M. (2012). The stat3/socs3a pathway is a key regulator of hair cell regeneration in zebrafish. [corrected]. J. Neurosci. *32*, 10662–10673. https://doi.org/10.1523/JNEUROSCI.5785-10.2012.

65. Schindelin, J., Arganda-Carreras, I., Frise, E., Kaynig, V., Longair, M., Pietzsch, T., Preibisch, S., Rueden, C., Saalfeld, S., Schmid, B., et al. (2012). Fiji: an open-source platform for biological-image analysis. Nat. Methods *9*, 676–682. https://doi.org/10.1038/nmeth.2019.

66. Oliveros, J.C. (2007-2015). An Interactive Tool for Comparing Lists with Venn's Diagrams. https://bioinfogp.cnb.csic.es/tools/venny/index.html.

67. Qiu, X., Mao, Q., Tang, Y., Wang, L., Chawla, R., Pliner, H.A., and Trapnell, C. (2017). Reversed graph embedding resolves complex single-cell trajectories. Nat. Methods *14*, 979–982. https://doi.org/10.1038/nmeth.4402.

68. Kent, W.J. (2002). BLAT–the BLAST-like alignment tool. Genome Res. *12*, 656–664. https://doi.org/10.1101/gr.229202.

69. Ashburner, M., Ball, C.A., Blake, J.A., Botstein, D., Butler, H., Cherry, J.M., Davis, A.P., Dolinski, K., Dwight, S.S., Eppig, J.T., et al. (2000). Gene ontology: tool for the unification of biology. The Gene Ontology Consortium. Nat. Genet. *25*, 25–29. https://doi.org/10.1038/75556.

70. Carbon, S., Ireland, A., Mungall, C.J., Shu, S., Marshall, B., and Lewis, S.AmiGO Hub; Web Presence Working Group (2009). AmiGO: online access to ontology and annotation data. Bioinformatics *25*, 288–289. https://doi.org/10.1093/bioinformatics/btn615.

71. Gene Ontology Consortium (2021). The Gene Ontology resource: enriching a GOld mine. Nucleic Acids Res. *49*, D325–D334. https://doi.org/10.1093/nar/gkaa1113.

72. Heinz, S., Benner, C., Spann, N., Bertolino, E., Lin, Y.C., Laslo, P., Cheng, J.X., Murre, C., Singh, H., and Glass, C.K. (2010). Simple combinations of lineage-determining transcription factors prime cis-regulatory elements required for macrophage and B cell identities. Mol. Cell *38*, 576–589. https://doi.org/10.1016/j.molcel.2010.05.004.

73. Quinlan, A.R., and Hall, I.M. (2010). BEDTools: a flexible suite of utilities for comparing genomic features. Bioinformatics *26*, 841–842. https://doi.org/10.1093/bioinformatics/btq033.

74. Quinlan, A.R. (2014). BEDTools: the Swiss-army tool for genome feature analysis. Curr. Protoc. Bioinformatics *47*. https://doi.org/10.1002/0471250953.bi1112s47.

75. Ganis, J.J., Hsia, N., Trompouki, E., de Jong, J.L.O., DiBiase, A., Lambert, J.S., Jia, Z., Sabo, P.J., Weaver, M., Sandstrom, R., et al. (2012). Zebrafish globin switching occurs in two developmental stages and is controlled by the LCR. Dev. Biol. *366*, 185–194. https://doi.org/10.1016/j.ydbio.2012.03.021.

76. Quillien, A., Abdalla, M., Yu, J., Ou, J., Zhu, L.J., and Lawson, N.D. (2017). Robust identification of developmentally active endothelial enhancers in zebrafish using FANS-assisted ATAC-seq. Cell Rep. *20*, 709–720. https://doi.org/10.1016/j.celrep.2017.06.070.

77. Mahony, S., and Benos, P.V. (2007). STAMP: a web tool for exploring DNA-binding motif similarities. Nucleic Acids Res. *35*, W253–W258. https://doi.org/10.1093/nar/gkm272.

## STAR★METHODS

### KEY RESOURCES TABLE

| REAGENT or RESOURCE | SOURCE | IDENTIFIER |
|---|---|---|
| **Antibodies** | | |
| Alexa Fluor 488 Phalloidin | Invitrogen | Cat#A12379 |
| Alexa Fluor 568 Goat anti-Rabbit IgG | Invitrogen | Cat#A11036; RRID: AB_10563566 |
| Sox2 antibody | GeneTex | Cat#GTX124477; RRID: AB_11178063 |
| **Chemicals, peptides, and recombinant proteins** | | |
| Diphtheria Toxin, from *Corynebacterium diphtheriae* | Sigma-Aldrich | D0564 |
| MS-222 | Western Chemical, Inc | TRSI; CAS: 886-86-2 |
| EnGen Cas9 NLS, *S. pyogenes* | New England Biolabs | Cat#M0646T |
| AmpliTaq Gold DNA Polymerase with Gold Buffer and MgCl$_2$ | Applied Biosystems | Cat#4311818 |
| GeneScan ROX400 size standard | Applied Biosystems | Cat#402985 |
| HiDi Formamide | Applied Biosystems | Cat#4311320; CAS: 75-12-7 |
| Phusion DNA Polymerase | New England Biolabs | Cat#M0530 |
| pCR2.1-TOPO cloning kit | Invitrogen | Cat#450641 |
| TRIzol Reagent | Invitrogen | Cat#15596026 |
| PowerUp SYBR Green Master Mix | Applied Biosystems | Cat#A25779 |
| SuperScript III Reverse Transcriptase | Invitrogen | Cat#18080044 |
| Copper (II) Sulfate | Sigma-Aldrich | Cat#451657; CAS: 7758-98-7 |
| YO-PRO-1 | Life Technologies | Cat#Y3603 |
| **Critical commercial assays** | | |
| Chromium Single Cell 3′ GEM, Library & Gel Bead Kit v3 | 10x Genomics | 1000092 |
| Chromium Chip B Single Cell Kit | 10x Genomics | 1000074 |
| Chromium Single Cell ATAC Library & Gel Bead Kit | 10x Genomics | 1000111 |
| Chromium i7 Multiplex Kit N Set A | 10x Genomics | 1000084 |
| Chromium Single Cell ATAC Chip E Kit | 10x Genomics | 1000086 |
| Bioanalyzer DNA High Sensitivity Kit | Agilent | 5067–4626 |
| Acridine Orange/Propidium Iodide (AO/PI) Cell Viability Kit | Logo Biosystems | LGBD10012 |
| NextSeq 500/550 High Output Kit v2.5 (150 cycles) 400 million reads | Illumina | 20024907 |
| **Deposited data** | | |
| Raw data | This paper | GEO: GSE192947 |
| Genome Reference Consortium Zebrafish Build 11 | Genome Reference Consortium | https://www.ncbi.nlm.nih.gov/grc/zebrafish |
| **Experimental models: Cell lines** | | |
| TAB5 zebrafish inner ear sensory epithelia, saccule and utricle | Burgess Lab | N/A |
| Tg(*myo6b*:DTR) zebrafish inner ear sensory epithelia, saccule and utricle | Burgess Lab | N/A |
| *sox2$^{hg138}$* zebrafish inner ear sensory epithelia, saccule and utricle | Burgess Lab | N/A |
| **Experimental models: Organisms/strains** | | |
| Tg(*myo6b*:hDTR) | Burgess Lab | N/A |
| sox2$^{hg138}$ | Burgess Lab | ZDB-ALT-220104-7 |
| sox2$^{hg139}$ | Burgess Lab | ZDB-ALT-220329-6 |
| sox2$^{hg140}$ | Burgess Lab | ZDB-ALT-220329-7 |
| TAB5 | Burgess Lab | N/A |

*(Continued on next page)*

*Continued*

| REAGENT or RESOURCE | SOURCE | IDENTIFIER |
|---|---|---|
| Oligonucleotides | | |
| See Table S17 for Oligonucleotides used in this study | N/A | N/A |
| Software and algorithms | | |
| 10x Genomics Cell Ranger 6.0.0 | 10x Genomics | https://support.10xgenomics.com/single-cell-gene-expression/software/overview/welcome |
| 10x Genomics Cell Ranger ATAC 2.0.0 | 10x Genomics | https://support.10xgenomics.com/single-cell-atac/software/overview/welcome |
| Seurat code for processing scRNA-seq data | Stuart et al., 2019; Butler et al., 2018 | https://satijalab.org/seurat/index.html |
| Signac code for processing scATAC-seq data | Stuart et al., 2019; Butler et al., 2018 | https://satijalab.org/signac/index.html |
| Monocle3 | Trapnell et al., 2014 | https://cole-trapnell-lab.github.io/monocle3/ |
| GREAT 3.0.0 | Hiller et al., 2013 | http://great.stanford.edu/great/public-3.0.0/html/ |
| LiftOver | Kent et al., 2002 | https://genome.ucsc.edu/cgi-bin/hgLiftOver |
| Prism 9 | GraphPad | https://www.graphpad.com |
| R/RStudio | RStudio Team, 2020 | https://www.rstudio.com |
| AMIGO 2 | M. Ashburner et al., 2000; S. Carbon et al., 2009; C. Gene Ontology, 2021 | http://amigo.geneontology.org/amigo |
| HOMER | Heinz et al., 2010 | http://homer.ucsd.edu/homer/ |
| BEDTools | A. R. Quinlan et al., 2010; A. R. Quinlan, 2014 | http://bedtools.readthedocs.io/en/latest/ |
| ImageJ/FIJI | J. Schindelin et al., 2012 | https://imagej.net/software/fiji/#downloads |
| Venny 2.1.0 | J. C. Oliveros, 2007–2015 | https://bioinfogp.cnb.csic.es/tools/venny/ |
| Other | | |
| Luna Automated Fluorescence Cell Counter | Logo Biosystems | LGBD10002 |
| NextSeq 550 Sequencing System | Illumina | https://www.illumina.com |
| Zeiss LSM 880 Confocal | Zeiss | https://www.zeiss.com/corporate/int/home.html |
| 35G Beveled Needle | World Precision Instruments | NF35BV-2 |
| Chromium Controller | 10x Genomics | https://www.10xgenomics.com/instruments/chromium-controller |
| ABI Genetic Analyzer 3130xl | Applied Biosystems | Cat#4359571 |
| QuantStudio 3 Real-Time PCR System | Applied Biosystems | Cat#A28567 |
| 2100 Bioanalyzer Instrument | Agilent | G2939B |
| 10 μL NanoFil microsyringe | World Precision Instruments | NANOFIL |

## RESOURCE AVAILABILITY

### Lead contact
Further information and requests for resources and reagents should be directed to and will be fulfilled by the lead contact, Shawn Burgess (burgess@mail.nih.gov).

### Materials availability
Requests for zebrafish lines generated in this study should be directed to and will be fulfilled by the lead contact.

### Data and code availability
Single-cell RNA-seq and single-cell ATAC-seq data is available on the NCBI Gene Expression Omnibus (GEO) with accession GSE192947. The accession number is also listed in the key resources table. All other data are available in the main text or the supplementary materials.

This paper does not report original code. DOIs for pre-existing code used in this paper is listed in the key resources table.

Any additional information required to reanalyze the data reported in this paper is available from the lead contact upon request.

## EXPERIMENTAL MODEL AND SUBJECT DETAILS

TAB5 (wild-type, WT), Tg(*myo6b*:hDTR), *sox2*^*hg138*^, *sox2*^*hg139*^, and *sox2*^*hg140*^ zebrafish used in this study were housed and raised on a recirculating aquaria system at the National Institutes of Health, using methods and parameters previously described.[55] For adult hair cell ablation studies, adult zebrafish were randomly selected and represented roughly equal numbers of males and females. As sex was not tracked during the experiments, potential effects stemming from sex differences could not be determined. All experiments were approved by the Institutional Animal Care and Use Committee for the National Human Genome Research Institute under Animal Study Protocol: G-01-3.

## METHOD DETAILS

### Adult zebrafish diphtheria toxin administration

Diphtheria toxin was purchased from Sigma-Aldrich and dissolved in 1X PBS. 6 to 10 month-old wild-type (TAB5) and transgenic adult zebrafish of mixed sex were given one intraperitoneal (IP) injection with diphtheria toxin using a 10 μL NanoFil™ microsyringe with a 35G beveled needle. Total protein injected into each fish was 0.05 ng in a total volume of 1 μL. Fish were fasted for 24 hours prior to IP injection. Buffered tricaine (<0.04 g/L, MS-222) diluted in aquaria water was used to immobilize the fish, they were then placed (inverted) into a cut sponge and given the IP injection into the abdominal cavity, posterior to the pelvic girdle. Immediately after injection, fish were recovered in fresh system water and maintained off system for up to 23 days at a maximum density of 5 fish/L. Fish were fed Gemma 300 at approximately 1.5% body weight daily. Water quality (pH, ammonia, nitrite, nitrate, temperature) was monitored twice daily. At least 50% of the water was changed daily. Health monitoring was performed twice daily and any fish that appeared to be in pain or distress were euthanized. On days 4, 5, and 7, adult zebrafish were randomly selected and represented roughly equal numbers of males and females. Inner ear tissues were harvested for profiling experiments and over a broader range for other experiments. All experiments were approved by the Institutional Animal Care and Use Committee for the National Human Genome Research Institute under Animal Study Protocol: G-01-3.

### Fine dissection of adult inner ear for single cell suspensions

For single-cell experiments, fresh inner ear sensory epithelia (saccule and utricle) were harvested from at least 35 adult zebrafish of mixed sexes on days 4, 5, and 7 post-DT injection. Adult zebrafish were euthanized with buffered MS-222 followed by decapitation. Inner ear organs were removed from heads of decapitated zebrafish as described.[56] Fine dissection of inner ear sensory epithelia was carried out in 1X PBS without calcium and magnesium. Saccule and utricle were separately processed for single-cell experiments.

### Single-cell suspensions on adult sensory epithelia

Dissected sensory epithelia were dissociated for single-cell experiments using a previously described protocol with 1% BSA used in place of FBS.[57] 10–15 pairs of sensory epithelia (saccule or utricle) were dissected, enzymatically digested and filtered, and then consolidated during the filtration step of the single-cell suspension preparation. Consolidated cell suspensions were then centrifuged for 5 minutes at 700 g at room temperature and resuspended in 20 μL of 1X PBS (without calcium and magnesium) with 0.04% BSA. Cell number and viability was determined using a Luna Automated Fluorescence Cell Counter and Acridine Orange/Propidium Iodide (AO/PI) Cell Viability Kit (Logo Biosystems). Cell viability ranged from 80%-95%.

### 10x genomics scRNA-seq library construction

For scRNA-seq, up to 7,000 cells per sample in 46.6 μL 1X PBS +0.04% BSA were loaded onto a Chromium Chip B (10x Genomics) and run using the Chromium Controller (10x Genomics) to generate single cell beads in the emulsion (GEM) according to manufacturer protocol (10x Genomics). cDNA libraries were generated with Chromium Single Cell 3′ GEM, Library and Gel Bead Kit V3 (10x Genomics) and Chromium i7 Multiplex Kit (10x Genomics). Quality control for the constructed libraries were performed using the Agilent Bioanalyzer High Sensitivity DNA kit (Agilent) and 2100 Bioanalyzer Instrument (Agilent). Libraries were sequenced using the NextSeq 500/550 High Output Kit v2.5 (150 cycles) 400 million reads (Illumina) on an Illumina NextSeq 550 System.

### 10x genomics scATAC-seq library construction

For scATAC-seq, nuclei were prepared from cell suspensions according to 10x Genomics Chromium Low Cell Input Nuclei Isolation preparation guidelines from fresh cells. Nuclear integrity and concentration were determined using a Luna Automated Fluorescence Cell Counter and Acridine Orange/Propidium Iodide (AO/PI) Cell Viability Kit (Logo Biosystems). Nuclei were adjusted to the desired capture number concentration based on the number of available nuclei. Up to 16,000 nuclei per sample were immediately incubated in a Transposition Mix to fragment DNA in open regions of chromatin and add adapter sequences to the ends of DNA fragments using the Chromium Single Cell ATAC Library & Gel Bead Kit (10x Genomics). GEMs were generated by combining barcoded Gel Beads, transposed nuclei, a Master Mix, and Partitioning Oil on a Chromium Chip E (10x Genomics) and run using the Chromium Controller (10x Genomics). Libraries were generated according to the manufacturers protocol with Primary Cell Total Cycles based on Targeted Nuclei Recovery. Quality control for the constructed libraries were performed using the Agilent Bioanalyzer High Sensitivity DNA kit

(Agilent) and 2100 Bioanalyzer Instrument (Agilent) to determine fragment size. Libraries were sequenced using the NextSeq 500/550 High Output Kit v2.5 (150 cycles) 400 million reads (Illumina) on an Illumina NextSeq 550 System.

### CRISPR/Cas9 enhancer deletion

CRISPR/Cas9 mutagenesis was performed as previously described.[58] To study the effect of the upstream enhancer of *sox2* deletion on regeneration, we implemented the strategy described in Goudarzi et al.[59] Cas9 protein (New England Biolabs, NEB) was co-injected with 6 guide RNAs (Eurofins) flanking the *sox2* upstream enhancer into the yolk of single cell-stage embryos. See Table S17 for 6 sgRNA's designed to flank the upstream enhancer to generate the deletion.

Mutations rates were determined by PCR amplification using a pair of external primers flanking the outermost guide RNA targets. PCR reactions used 2 μL of diluted DNA and 5 μL of PCR mix containing AmpliTaq Gold DNA Polymerase (Applied Biosystems) with appropriate buffer, MgCl$_2$, dNTPs, and equimolar ratios of the following three primers at 5 pmol/μL: M13F primer with fluorescent tag (6-FAM), external amplicon-specific forward primer (AATGCGTGAATAAGCCGAAT) with M13 forward tail (5′-TGTAAAACGACGGC CAGT-3′) and 5′ PIG-tailed (5′-GTGTCTT-3′) amplicon-specific reverse primer (TTTATGGCAGCGGGCTATAC).[60]

PCR conditions were as follows: denaturation at 98°C for 10 min, followed by 35 cycles of amplification (98°C for 30 sec, 55°C for 30 sec, and 72°C for 1 min), a final extension at 72°C for 10 min, and indefinite hold at 4°C. 10 μL of 1:25 mixture of ROX400 size standard and Hi-Di formamide (Applied Biosystems) were added to 2 μL of PCR product and samples were denatures at 95°C for 5 min. Denatured PCR products were analyzed to identify wild-type and mutant fragments generated by deletion on a Genetic Analyzer 3130xl using POP-7 polymer. Data were analyzed for allele sizes and corresponding peak heights using the local Southern algorithm available in the Genescan and Genotyper software of GeneMapper software package (Applied Biosystems).

Injected embryos were raised to generate founder zebrafish. Founder fish were then outcrossed to wild-type (TAB5) to generate heterozygous F$_1$ zebrafish. Siblings carrying enhancer deletions were then crossed to generate F$_2$ progeny and phenotype-genotype correlations were done using the F$_2$ embryos or adults. Enhancer deletion mutant zebrafish were identified by fluorescence PCR and sequencing using DNA extracted from fin clips. PCR was performed using a pair of primers flanking the outermost guide RNA targets and a pair of primers with one oligo specific to the internal targeted region.

To screen for enhancer deletion mutants, we performed two PCR reactions per sample. The first PCR reaction included two external primers that flank the deleted region and a second PCR reaction that included a primer that specific to an internal site where the deleted region is located and an external primer. External amplicon-specific forward primer (AATGCGTGAATAAGCCGAAT) with M13 forward tail (5′-TGTAAAACGACGGCCAGT-3′) and 5′ PIG-tailed (5′-GTGTCTT-3′) external amplicon-specific reverse primer (TTTATGGCAGCGGGCTATAC). External amplicon-specific forward primer (AATGCGTGAATAAGCCGAAT) with M13 forward tail (5′-TGTAAAACGACGGCCAGT-3′) and 5′ PIG-tailed (5′-GTGTCTT-3′) internal amplicon-specific reverse primer (TGACAACAGCC GAAACAAAA). In the first PCR reaction with two external primers, amplification from the wild-type template results in the production of one full-length 2811 bp fragment. Amplification from the deleted mutant (*sox2*[hg138]) results in one short fragment ~265 bp (and one full-length fragment if heterozygous). In the second PCR reaction with a primer that is specific to an internal site where the deleted region is located and an external pair, amplification from the wild-type template results in production of a short fragment 275 bp. Amplification from the deletion mutant does not result in any fragments (unless mutants are heterozygous). Three independent alleles were identified.

To identify the molecular lesion of the CRISPR-induced deletion, PCR products were amplified from DNA of F$_1$ zebrafish containing heterozygous deletions from three independent alleles: *sox2*[hg138], *sox2*[hg139], *sox2*[hg140]. PCR reactions used 2 μL of diluted DNA and 5 μL of PCR mix containing Phusion DNA Polymerase (NEB) with 5X Phusion HF buffer, dNTPs, and equimolar ratios of the following primers at 10 μM: Forward – AATGCGTGAATAAGCCGAAT and Reverse – TTTATGGCAGCGGGCTATAC. PCR products were purified (MinElute PCR Purification Kit, Qiagen) and subcloned into pCR2.1-TOPO (Invitrogen) vector by TA cloning according to the manufacturer's instructions. Following bacterial transformation, 4 colonies were picked and grown overnight for each allele. Plasmid DNA was extracted (Qiagen Miniprep Kit) and analyzed by Sanger sequencing.

### RNA isolation from adult inner ear tissues

Adult inner ear sensory epithelia (saccule) were dissected from 6-8 adult zebrafish of mixed sex from untreated genotypes and from DT injected zebrafish on days 4, 5, 7, 9, and 11 from TAB5 (wild-type, WT), heterozygous Tg(*myo6b*:hDTR), and heterozygous *sox2*[hg138]/Tg(*myo6b*:hDTR) zebrafish. Adult sensory epithelia were homogenized in 0.7 mL TRIzol Reagent (Invitrogen) with a power homogenizer. RNA was isolated from the aqueous phase after TRIzol/chloroform extraction and treated with DNase I. RNA was purified using the RNA Clean & Concentrator-5 (Zymo Research) and measured (Nanodrop One).

### Quantitative real-time PCR analysis on adult inner ear tissues

RNA was transcribed into cDNA according to manufacturer's instructions (SuperScript III RT, Invitrogen). Quantitative Real-Time PCR Analysis (RT-qPCR) was performed in technical replicates using 1:5 cDNA in each reaction and a primer concentration of 0.5 μM. PowerUp SYBR Green Master Mix (Applied Biosystems) and self-designed primers were used (Eurofins). The RT-qPCR reaction was completed on the QuantStudio 3 Real-Time PCR System (Applied Biosystems) with the following cycling conditions: 95°C for 10 min, and 40 cycles of 95°C for 15 sec, 60°C for 1 min. Primers were designed by using Primer3 followed by a UCSC *in silico* PCR to search the zebrafish sequence database. *sox2* was amplified using the forward primer 5′- ACTCCATGACCAACTCGCAG-3′ and

the reverse primer 5′- AATGAGACGACGACGTGACC-3′. *ef1alpha* was used as a housekeeping gene and was amplified using the forward primer 5′-CGACAAGAGAACCATCGAGAAGTT-3′ and the reverse primer 5′-CCAGGCGTACTTGAAGGA-3′. qPCR Ct values were analyzed using the double delta Ct method.[61]

### Lateral line hair cell and neuromast quantification

Hair cell staining and quantification were performed as described[62] using YO-PRO-1 (Life Technologies). For hair cell regeneration analysis, embryos from wild-type (TAB5) and from heterozygotic F$_2$ *sox2$^{hg138}$, sox2$^{hg139}$, and sox2$^{hg140}$* in-crosses at 5 dpf were treated with copper (II) sulfate (CuSO$_4$) (Sigma-Aldrich) at 10 μM for 1 h at 28.5°C, recovered for 48 h at 28.5°C, and then counted for the regenerated hair cells in the lateral line neuromasts P1, P2, P4, and P5. Larvae were then genotyped to detect wild-type and enhancer deletion alleles. Approximately 40 embryos were used for each of the analyses. A two-way ANOVA comparison of the data obtained on untreated wild-type larvae with the data obtained from untreated enhancer deletion mutant larvae was performed using Prism 9 (GraphPad). A two-way ANOVA comparison of the data obtained on untreated larvae with the data obtained from CuSO$_4$ treated larvae was performed using Prism 9 (GraphPad). p-value was determined by ANOVA and Sidak multiple comparison test.

### Histological methods for imaging

Adult zebrafish were euthanized using buffered MS-222 followed by decapitation. The heads were dissected and fixed in 4% formaldehyde overnight at 4°C. Inner ears were gross dissected in 1X PBS as previously described in Liang and Burgess.[63] The dissection techniques were followed exactly as described.

### Adult inner ear hair cell labelling

Alexa Fluor 488 phalloidin (Invitrogen) was used to visualize and quantify F-actin in stereocilia of zebrafish inner ear sensory epithelia. Fixed saccule and utricle were stained using Alexa Fluor 488 phalloidin for 20 minutes at room temperature as previously described in.[63,64] Proteins were detected in whole-mount utricles and saccules using standard immunofluorescence labeling methods. Primary and secondary antibodies used include the rabbit Sox2 (GeneTeX, GTX124477, 1:300-dilution) and Alexa Fluor 568 goat anti-rabbit IgG (Invitrogen, 1:1000-dilution), respectively.

### Cellular imaging and analysis of inner ear hair cells

Confocal images were acquired with a Zeiss LSM 880 confocal microscope (Zeiss). Confocal Z stacks of the entire saccule and utricle were projected into a single image to capture all phalloidin positive cells from different planes of focus for counting. Quantification was performed by selecting 2.5 mm$^2$ from confocal images of whole mounts and phalloidin positive cells were counted with Image J/Fiji software.[65] ANOVA comparisons were performed in Prism 9 (Graphpad). 4-8 samples in each group were used for statistical analysis. p-value was determined by ANOVA and Sidak multiple comparison test.

## QUANTIFICATION AND STATISTICAL ANALYSIS

### scRNA-seq data processing

We used the Cell Ranger (10x Genomics, v6.0.0) analysis pipeline to process Chromium single-cell data to align reads, generate feature-barcode matrices, and other secondary analysis.

Using Cell Ranger, Illumina base call files (BCL) were demultiplexed with the cellranger mkfastq to generate FASTQ files. The FASTQ files were aligned to the *Danio rerio* genome reference sequence (danRer11) and filtered followed by barcode and UMI counting using cellranger count. Cellranger generated raw and filtered feature-barcode matrices. All raw scRNA-seq data was deposited to GEO under accession GSE192947.

### scRNA-seq analysis with Seurat

Filtered feature-barcode matrices were loaded into R Studio (v4.0) and analyzed with the Seurat R package.[24,28] Cells that had unique feature counts over 1000 or less than 200 and over 5% mitochondrial counts were filtered. After filtering, we used Seurat's NormalizeData function with the global-scaling normalization method "LogNormalize" to normalize the feature expression measurements for each cell by the total expression, multiplied by the default scale factor (10,000) followed by log transformation. Highly variable features were identified using the FindVariableFeatures function (selection.method = "vst") followed by linear transformation using the ScaleData function on all genes. Principal component analysis (PCA) was performed on the scaled data. The dimensionality of the dataset was determined by exploring principal components (PCs) to determine the number of PCs to use for clustering. Cell clusters were identified using Seurat function FindClusters and previously defined dimensionality of the dataset (first 10 PCs). UMAP dimensional reduction was performed on scaled data using Seurat function RunUMAP. Differential expression (DE) testing was performed using Seurat to find markers that define clusters and perform pairwise cluster comparisons. Seurat performed differential expression based on the non-parametric Wilcoxon rank sum test. Markers to define every cluster compared to all remaining cells were identified using the Seurat function FindAllMarkers. The FindMarkers function was used to find markers distinguishing clusters during a pairwise comparison.

### Cell numbers in cell populations of homeostatic and regenerating sensory epithelia

To examine the dynamics of individual cell populations during hair cell regeneration we assessed the number of cells in each cluster present in regenerating sensory epithelia compared to homeostatic or non-regenerating controls. First, homeostatic or non-regenerating sensory epithelia from untreated wild-type, day 4 post-DT treated wild-type, and untreated Tg(*myo6b*:hDTR) transgenic fish were integrated via Seurat with regenerating sensory epithelia from Tg(*myo6b*:hDTR) transgenic fish treated with DT (days 4, 5, 7) to promote the identification of common cell types and enable comparative analysis. We randomly down-sampled the integrated object to 7000 cells in R: downsampled.obj <- large.obj[, sample(colnames(large.obj), size =7000, replace = F)

Cells were split into a stimulated group (stim = regenerating) and control group (non-regenerating). The stimulated group (stim = regenerating) consisted of 3,330 single cells. The control group (non-regenerating) consisted of 3,670 single cells.

The number of cells in each cluster was determined in R: table(Idents(downsampled.obj), downsampled.obj$stim)

### Pairwise comparisons and DE testing using Seurat

For inter-organ comparisons (saccule vs. utricle), larval neuromast and adult inner ear comparisons, and pairwise comparisons to identify regeneration response genes in regenerating tissues (Untreated vs. Day 4, Untreated vs. Day 5, Untreated vs. Day 7), we followed Seurat's 'Integrating stimulated vs. control PBMC datasets to learn cell-type specific responses' vignette (https://satijalab.org/seurat/archive/v3.0/immune_alignment.html).

Seurat objects were created with raw Cell Ranger generated gene expression matrices with the CreateSeuratObject function. Features detected in at least 5 cells (min.cells = 5) and cells where at least 500 features (min.features = 200) were analyzed. After filtering, we used Seurat's NormalizeData function with the global-scaling normalization method "LogNormalize" to normalize the feature expression measurements for each cell by the total expression, multiplied by the default scale factor followed by log transformation. Highly variable features were identified using the FindVariableFeatures function with selection.method = "vst". To integrate the data (or align), anchors were identified using the FindIntegrationAnchors function and the two datasets were integrated together with IntegrateData (dims = 1:20). A single integrated analysis was performed on all cells. Linear transformation on all cells was performed using the ScaleData function on all genes. Principal component analysis (PCA) was performed on the scaled data. The dimensionality of the dataset was determined by exploring PCs to determine the number of PCs to use for clustering. Cell clusters were identified using the Seurat function FindClusters.

Following integration, conserved cell type markers were identified using the FindConservedMarkers() function in each cluster. Support cell (SC), progenitor cell (PC), and hair cell (HC) clusters were annotated according to marker genes. To identify differentially expressed genes between two datasets or conditions for cells of the same cluster cell type, we used the FindMarkers() function using the options ident.1 = "cluster_of_interest_dataset_1", ident.2 = "cluster_of_interest_dataset_2". This command was run for all hair cell lineage clusters, the SC, PC, and HC. Seurat performed differential expression based on the non-parametric Wilcoxon rank sum test via the FindMarkers function.

For inter-organ comparisons, differentially expressed genes between the two datasets was performed for SC, PC, and HC types using the FindMarkers function. For example, to find hair cell specific gene expression differences between non-regenerating saccule and utricle, we used the FindMarkers() function using the options ident.1 = "HC_cluster_non_regenerating_saccule", ident.2 = "HC_cluster_non_regenerating_utricle". Similarly, for larval neuromast and adult inner ear comparisons, we used the FindMarkers() function using the options ident.1 = "HC_cluster_larval_neuromast", ident.2 = "HC_cluster_adult_inner_ear". The FindMarkers function was run for all hair cell lineage clusters, the SC, PC, and HC.

To identify regeneration response genes in regenerating tissues, we performed pairwise comparisons between non-regenerating and regenerating datasets: Untreated DTR Utricle vs. Day 4 DTR Utricle, Untreated DTR Utricle vs. Day 5 DTR Utricle, Untreated DTR Utricle vs. Day 7 DTR Utricle, Untreated DTR Saccule vs. Day 4 DTR Saccule, Untreated DTR Saccule vs. Day 5 DTR Saccule, Untreated DTR Saccule vs. Day 7 DTR Saccule. To identify differentially expressed genes between two conditions for cells of the same cluster, we used the FindMarkers() function using the options ident.1 = "cluster_of_interest_dataset_1", ident.2 = "cluster_of_interest_dataset_2". For example, FindMarkers() function using the options ident.1 = "HC_cluster_Untreated_DTR_Saccule", ident.2 = "HC_cluster_Day4_DTR_Saccule". This command was run for all hair cell lineage clusters, the SC, PC, and HC. Differential genes used for downstream analysis were required to have a p-value < 0.1 and an average log fold change of 0.25 in at least one dataset. Intersection or overlap between the three gene lists was performed using Venny 2.1.0.[66]

### Pseudotime analysis using Monocle 3

To construct single cell trajectories, Monocle 3 was used to cluster cells and reduce the dimensionality of gene expression matrices using UMAP.[23,32,56,67] We followed the Monocle 3 package documentation (https://cole-trapnell-lab.github.io/monocle3/). The scRNA-seq was first normalized with a specified number of principal components (num_dim = 100). The data was projected into a low-dimensional space with UMAP using the reduce_dimension() function in Monocle 3. Subsets of cells that lead to the hair cell lineage were chosen using the choose_cells() function for downstream analysis. A principle graph was fit on the subset of cells using the learn_graph() function and the cells were ordered in pseudotime according to their progress through the regeneration program using the order_cells() function. Cells were re-clustered in Monocle 3 for pseudotime and differential expression analysis (resolution = $1 \times e^{-2}$ for louvain clustering). The clusters identified as support cells furthest from the hair cells were chosen as "roots" of the trajectory.

### Finding modules of co-regulated genes using Monocle 3

To perform differential expression analysis to find modules of co-regulated genes, we used Monocle 3. We followed the Monocle 3 documentation (https://cole-trapnell-lab.github.io/monocle3/docs/differential/). We identified sets of genes that varied across clusters and Monocle 3 grouped them into modules according to similar patterns of expression. To identify genes whose expression changed significantly, we applied the Monocle 3 graph test() function that uses a statistic from spatial autocorrelation analysis called Moran's I (q-value < 0.05). Monocle 3 grouped genes that vary across clusters into modules using the find_gene_modules() function with resolution = $1 \times e^{-3}$ which runs UMAP on the genes and groups them into modules using Louvian community analysis.

### scATAC-seq data processing

We used the Cell Ranger ATAC (10x Genomics, v2.0.0) analysis pipeline to process Chromium Single Cell ATAC data to demultiplex raw base call (BCL) files generated by Illumina sequencing into FASTQ files. The FASTQ files were aligned to the *Danio rerio* genome reference sequence (danRer11/GRCz11) and filtered followed by barcode counting and peak calling using cellranger-atac count. Cell Ranger generated raw and filtered feature-peak matrices. All data was deposited to GEO under accession GSE192947.

### scATAC-seq analysis with Signac and Seurat

Using the Signac (v1.4.0) and Seurat R packages, filtered feature-peak matrices (chromatin data) and cell metadata generated by cellranger-atac was pre-processed and a Suerat object was created by following a pre-existing tutorial (https://satijalab.org/signac/articles/pbmc_vignette.html).[24,28] Using Signac, QC metrics for scATAC-seq experiments were computed by following the metrics outlined in the Analyzing PBMC scATAC-seq workflow.[24] The nucleosome binding pattern, TSS enrichment score, total number of fragments in peaks, and fraction of fragments in peaks were inspected. Cells that were outliers for QC metrics were removed (peak_region_fragments > 2000 and < 30000, TSS.enrichment > 2, and blacklist_ratio < 0.05, nucleosome_signal < 4, pct_reads_in_peaks > 15).

We used Signac to perform term frequency-inverse document frequency (TF-IDF) normalization. TF-IDF normalization normalizes across cells to correct for differences in sequencing depth and across peaks to give higher values to rare peaks. For feature selection, we used the FindTopFeatures function to use all features (min.cutoff = 'q0') for dimensional reduction due to the dynamic range of the scATAC-seq data. Singular value decomposition (SVD) was performed on the TF-IDF matrix using all features (peaks) for linear dimensional reduction. Each LSI component and sequencing depth was assessed to determine if there was a strong correlation between the initial LSI component and the total number of cell counts. There was a strong correlation between the first LSI component and the total number of counts for the cell, so we performed downstream steps without the first LSI component. Cells were embedded in a low dimensional space and graph-based clustering and non-linear dimension reduction for visualization was performed according to Signac and Seurat R packages.

UMAP dimensional reduction was performed using Seurat function RunUMAP (dims:2:30). Cell clusters were identified using Seurat FindNeighbords and FindClusters functions. A gene activity matrix was created from peak matrix data and the zebrafish annotation (danRer11/GRCz11). All peaks that fell within gene bodies or 2 kb upstream of a gene were considered to facilitate cluster annotation by examining activity of cell-type-specific marker gene activity. The activities of canonical support cell (SC), progenitor cell (PC), and hair cell (HC) marker genes were visualized.

### scRNA-seq and scATAC-seq integration

To integrate the two modalities (scRNA-seq and scATAC-seq), we followed a pre-existing tutorial (https://satijalab.org/seurat/articles/integration_introduction.html and https://satijalab.org/seurat/archive/v3.0/atacseq_integration_vignette.html). Pre-processed scRNA-seq datasets were loaded using the methods for cross-modality integration and label transfer to confirm that scRNA-based classifications were consistent with the scATAC-seq data. Anchors between scATAC-seq and scRNA-seq datasets were identified and used to transfer cell type labels learned from scRNA-seq data to the scATAC-seq cells. For visualization purposes, scRNA-seq and scATAC-seq were co-embedded in the same low dimensional space and merged followed by UMAP analysis to visualize the cells together.

### Identification of RREs

Differentially expressed peaks were first identified in scATAC-seq datasets. Since Signac is an extension of Seurat, we used the FindAllMarkers function to obtain differentially expressed peaks for each of the clusters in a dataset with a min.pct of 0.025. Cluster assignments were made based on quantitative accessibility and gene expression. Differential peaks identified were required to have a p-value < 0.1 and an average log fold change enrichment of 0.25.

To identify regeneration response elements or RREs, we manually performed pairwise comparisons between non-regenerating (untreated) controls and regenerating datasets: Untreated DTR Utricle vs. Day 4 DTR Utricle, Untreated DTR Utricle vs. Day 5 DTR Utricle, Untreated DTR Utricle vs. Day 7 DTR Utricle, Untreated DTR Saccule vs. Day 4 DTR Saccule, Untreated DTR Saccule vs. Day 5 DTR Saccule, Untreated DTR Saccule vs. Day 7 DTR Saccule. DTR indicates the presence of the Tg(*myo6b*:hDTR) transgene. To identify shared or overlapping peaks between untreated and treated datasets, we used HOMER mergePeaks –d 100 and identified overlapping and unique peaks for each dataset in a cell type specific manner. We removed common peaks shared or

overlapping between untreated or homeostatic and treated samples leaving only the cell-type-specific emerging peaks. Cell type specific emerging peaks or RREs found in saccule and utricle were combined for downstream analysis.

The RRE peak data was used as input for downstream analysis including GREAT analysis, motif analysis, peak-to-gene-annotation and deep learning experiments.

### GREAT analysis and gene ontology annotations

GREAT version 3.0.0 (http://great.stanford.edu/great/public-3.0.0/html/) was used with the zebrafish assembly (danRer7, Jul/2010).[35] Zebrafish enhancer genome coordinates were converted from danRer11 genomic coordinates to danRer7 coordinates for the analysis (liftOver[68]). Functional annotations of zebrafish enhancers were performed with GREAT version 3.0.0 [35] using the basal plus extension mode and default parameters (5 kb upstream, 1 kb downstream, plus distal up to 1000 kb). GO enrichment analysis was performed using AmiGO 2.[69–71]

### Motif analysis using HOMER

To identify peak coordinates of interest containing instances of Sox and/or Six TF motifs, the annotatePeaks.pl function of HOMER with the "-m <motif file>" option was used.[72] Motif file outputs by HOMER were selected based on initial *de novo* motif enrichment using the findMotifsGenome.pl function. For analyzing genomic motifs, the danRer11 genome reference sequence was configured for use with HOMER.

### Peak-to-gene annotation

To differentiate accessible regions in the core promoter from those representing putative enhancers, we classified scATAC-seq peaks according to their position relative to the TSS of genes. After classifying regions that become more accessible during regeneration, we looked for regions within a 50 kb genomic window. Peaks were annotated by genes using bedtools window function and the GRCz11 assembly (release 96) annotation file in gtf format (Danio_rerio.GRCz11.96.gtf).[73,74] Usage of bedtools window is as follows:

```
bedtools window -a /Danio_rerio.GRCz11.96.gtf -b /cell_specific_RRE_genomic_coordinates.bed -l 50000 -r 50000 -sw
```

### Hypergeometric test

The overlapping probability between RRE peaks linked to DE genes or RRE peaks with CNEs linked to DE genes was estimated by using the hypergeometric distribution using the phyper function in R: phyper(min(n1,n2),n1, n-n1,n2) - phyper(m-1,n1,n-n1,n2), where n is the total number of genes in the zebrafish genome (GRCz11 coding genes: 25,592), n1 is the number of genes in one list, n2 is the number of genes in the second list, and m is the number of overlapping elements in the given lists.

### Identification of co-occurrence of *de novo* motifs using deep learning (DL)

To discover *de novo* binding motifs, we built enhancer DL models based on scATAC-seq emerging peaks in progenitor cells, hair cells and support cells from regenerating inner ears (saccule and utricle). The positive set contained 18,845, 17,948, and 14,050 regulatory elements in each cell type, respectively. Each element was defined as a 400 bp region centered at the ATAC-seq emerging peak in the corresponding cell type. The control set contained 400 bp elements centered at the open chromatin regions from endothelial, liver and red blood cells in zebrafish.[75,76] The training data set contained elements in positive set from all chromosomes, except for chromosomes 8 and 9, and 10-fold control sequences. The testing data set contained positive elements from chromosome 8 and 9 and 1-fold control sequences.

We constructed the DL model with four convolutional neural networks (CNNs) and detailed parameters can be found in Table S15. In the first CNN layer we used a sliding window of 9 bp (9-mer) and a step size of 3 bp to scan the input DNA sequences for *de novo* motifs, excluding the 50 bp regions from both ends to avoid potential errors near boundaries. We limited the maximum number of identified *de novo* motifs to 64 in order to reduce redundancy. For each *de novo motif*, we aligned all found 9-mer together and derived a position weight matrix (PWM). We next applied STAMP[77] to find the best-matching known TF binding motifs for each derived PWM with a p-value $< 1 \times e^{-5}$.

To identify the possible co-occurrence between any pair of *de novo* motifs in a range of 100 bp in DNA sequences, we retrieved the frequency that both *de novo* motifs were detected in the same filter in the third CNN layers. If the frequency is significantly higher (p-value < 0.001) than background, the two *de novo* motifs were considered to be co-occurrences.

