## [Document S2. Transparent peer review records for Jimenez et al. · Cell Genomics]

A regulatory network of Sox and Six transcription factors initiate a cell fate transformation during hearing regeneration in adult zebrafish

Author list

Erin Jimenez¹, Claire C. Slevin¹, Wei Song⁴, Zelin Chen^{2,3}, Stephen C. Frederickson¹, Derek Gildea¹, Weiwei Wu⁵, Abdel G. Elkahoul⁶, Ivan Ovcharenko⁴, Shawn M. Burgess^{1,5}

¹Translational and Functional Genomics Branch, National Human Genome Research Institute, Bethesda, MD, USA.

²CAS Key Laboratory of Tropical Marine Bio-Resources and Ecology, South China Sea Institute of Oceanology, Chinese Academy of Sciences, Guangzhou, China

³Southern Marine Science and Engineering Guangdong Laboratory (Guangzhou), China

⁴Computational Biology Branch, National Center for Biotechnology Information, National Library of Medicine, National Institutes of Health, Bethesda, Maryland 20894.

⁵Vaccine Immunology Program, Vaccine Research Center, National Institute of Allergy and Infectious Diseases, Bethesda, MD, USA ⁶Cancer Genetics and Comparative Genomics Branch, National Human Genome Research Institute, Bethesda, MD, USA.

Summary

Initial submission: Received : February 14th 2022

Scientific editor: Judith Nicholson

First round of review: Number of reviewers: 3
Revision invited : March 9th 2022
Revision received : May 12th 2022

Second round of review: Number of reviewers: 3
Accepted : Yes

Data freely available: Yes

Code freely available: Yes

This transparent peer review record is not systematically proofread, type-set, or edited. Special characters, formatting, and equations may fail to render properly. Standard procedural text within the editor's letters has been deleted for the sake of brevity, but all official correspondence specific to the manuscript has been preserved.

Referees' reports, first round of review

Reviewer 1:

This manuscript by Jimenez et al. is an elegant approach to identify gene expression and cis-regulatory changes accompanying regeneration of mechanosensory hair cells in the zebrafish inner ear. Single-cell transcriptome and chromatin accessibility are assayed during hair cell regeneration following genetic ablation in adults. Strengths of the study include looking directly at the inner ear rather than the more commonly studied lateral line that has less homology to the mammalian inner ear. A large number of regeneration-specific enhancers are also identified, and interestingly many of these show sequence conservation across fish species, suggestive of an ancient regulatory program for hair cell regeneration. The most striking aspect of the study is the identification of a regeneration-specific enhancer for *sox2*, which when deleted affects the timing of *sox2* expression and long-term hair cell regeneration. This is quite a surprising result given that knockout of regeneration-specific enhancers by other groups have not shown regeneration defects, that heterozygous *sox2* enhancer deletion has the same regeneration effect as homozygous deletion, and that enhancer deletion results in only subtle shifts in the timing of *sox2* expression. However, characterization of this enhancer deletion, including possible off-target effects, is not adequately discussed. If further data can be generated to support specificity of this enhancer deletion, including analysis of independent alleles, then this study will be of considerable interest to not only the hearing research community but the stem cell and genomics communities overall.

1. The exact sequence of the *sox2* enhancer deletion should be shown. It is a bit surprising that heterozygotes show the same defect as homozygotes. Were more than one independent deletion allele analyzed? CRISPR is known to cause large inversions and other chromosomal abnormalities that may be difficult to detect by PCR, as well as off-target lesions. The result should therefore be validated with at least two independent alleles, extensive outcrossing should be

performed to show the phenotype segregates with the intended deletion, and Southern blot should be used to rule out local chromosomal rearrangements (though similar phenotype in independent alleles would make this unnecessary).

2. In Fig 9C,D and Fig S6, it is unclear whether hets and/or homos are being examined, in contrast to the text which states both are being examined.

3. In the Discussion, a definitive model is put forth where SCs become PCs that enter mitosis, and then either become HCs or revert back to SCs. This can only be inferred from pseudotime analysis alone and should be greatly softened. Only definitive *in vivo* lineage tracing (and proliferation assays) can be used to confirm these bioinformatic predictions.

4. In the Abstract, the term "pluripotent" should be reserved for cells that can generate all cell types of the embryo. PCs in the inner ear are multipotent, i.e. generating hair and support cells only. Along these lines, the comment that "Sox genes in general are associated with pluripotent cells" is irrelevant and should be deleted.

5. Differences between the adult inner ear and larval lateral line could either be due to the different stages (adult/larvae) or the type of mechanosensory organ. Conclusions should be toned down in the absence of stage-matched comparisons. Also analysis in Fig S1 is rather cursory.

6. The switching on of *sox2* expression during regeneration in Fig S4 is quite subtle. Other *sox* genes also appear to switch on in a similar manner. Why is there no genetic redundancy of *sox* factors? This should be discussed.

7. Fig 8. For the majority of genes shown, upregulation during regeneration is not apparent. It would also be helpful to annotate cluster identity.

Minor comments:

1. Define "S" and "U" abbreviations in Fig 2 legend.
2. Hard to appreciate expression score of Module 3 in mature HCs in Fig 4D.
3. The method used to integrate scATAC and scRNA should be explained in main text as this was not a multiome experiment. Discussion of this integration in Methods is insufficient. Was an algorithm such as SNAP-ATAC used?
4. It would be helpful to annotate the clusters in Fig 5D. Also the dot plots in Fig 5E-G are not easy to interpret - many are very low expression and do not appear localized to specific regions as stated in text.
5. ZF/GF/C and CNE should be defined in legend to Fig 6
6. The discussion of machine learning to find enriched motifs in enhancers is misleading, as from the Methods it appears that this is just the Homer algorithm.
7. Fig 7A SC, it is confusing why a lack of link between SOX10/SOX9 and SOX10/SOX9 is shown, whereas links between SOX and SIX are show for PC and HC.
8. Fig 9. Panel E model is not described in legend. The dots panels are therefore hard to interpret.
9. The sox2 deleted element should be indicated in Fig 6C
10. There appear to be 3 CNE in Fig S5, not only 2 as described in text.

Reviewer 2:

In this manuscript Jimenez at al present a "tour de force" of single cell and ATAC-seq data on isolated cells from the adult inner ear cells and larval lateral line cells from the zebrafish. This is the first time such a careful comprehensive comparative analysis has been carried out and there is an amazing amount of new data presented in this manuscript which opens up many new avenues of

research. The authors have done a commendable job of using AI to extract really meaningful biologically relevant data from these huge datasets. From this they have then zoned in and identified a sox2 enhancer element which appears to be specific to hair cells regeneration, they present elegant CRISPR knock-out data showing how this mainly conserved enhancer is essential for normal hair cell regeneration. In theory the paper could be published as is as it is very suited to a journal focusing on Genomics. A few minor revisions would broaden the impact of the work:

1. Figure 1 and 9 need "n =x" for all biological experiments added to the figure legend.
2. The authors refer in the manuscript "differentiated Sox2 positive support cells are left undamaged". Sox2 is a marker of neural stem cells, so it seems the work "differentiated" is not appropriate in this context and should be removed.
3. Figure 9 shows interesting data to suggest that when the sox2 enhancer is deleted this results in a delay in the regeneration process, is this due to a delay in initiation of the regeneration response due to a delay in proliferation of the support cells in response to injury?
4. Does repeated hair cell ablation over a period of weeks/months in the CRISPR sox2 enhancer knock-out animals lead to a more striking regeneration defect, is the enhancer element needed to maintain and replenish the support cells?

Reviewer 3:

A regulatory network of Sox and Six transcription factors initiates a cell fate transformation during hearing regeneration in adult zebrafish. The research described in this manuscript is a tour de force and is worthy of publication in this journal. However, the actual writing is something of a surprise. I would suggest that the authors take another look and make some adjustments in particular the short introduction needs to explain zebrafish hair cell biology better to make the paper more accessible to a genomic audience. For such a magnitude of results, the discussion is also underwhelming. Everything mentioned below is a minor point – but this was not an exhaustive list.

Results: "L injection of diphtheria toxin (DT) leads to complete hair cell loss of auditory (sacculle) and vestibular (utricle) hair cells 5 days-post injection leaving a population of differentiated sox2 positive supporting cells undamaged (Jimenez et al., 2021) (Figure 1A)." Looking at both the figure and their other paper this is an

unnecessary over statement. You can see a few green cells at every stage. For this work 100% ablation was not required – reword.

Figure 1 A diagram of what we are looking at would make it easier for those not familiar with zebrafish ear anatomy.

Figure 1b There are no stats or “n” described for the experiment done. Why SEM not SD? “We dissected out sacculles and utricles at three time points: Days 4, 5 and 7 post-DT, and each were processed separately in both scRNA-seq and scATAC-seq experiments.” This anatomy has yet to be described or significance mentioned (although it is addressed better later in the text- add to intro Figure 1c - repeats some data from 1a and adds info for utricle. However, the resolution is now much lower and doesn't really add much. Perhaps 1c should just be the new info for utricle and at higher res.

Major plus that two negative controls are used in ablation – controlling for both DT and the transgene (presumably the expression of a human receptor) - it would be great if the premise were explained better and more info given when each control is used and why.

“We generated twelve transcriptomic profiles on inner ear tissue using the 10X Chromium system for droplet-based single-cell 3' RNA sequencing (scRNA-seq) and quantified each data set using the Cell Ranger 6.0.0 pipeline (10x Genomics).” What is meant by 12 transcriptomic profiles? 12 replicate experiments? 12 different conditions?

“Having established the different support, progenitor, and hair cell populations of the inner ear allowed us to interrogate the dynamic expression pattern of specific genes of interest in nonregenerating and regenerating inner ear tissues.” It is stated that you have established these conditions in non-regenerating and regenerating inner ear tissues. This was not done very clearly.

The data presented in figure 2 combines homeostatic scRNAseq data with regeneration data. It is not clear why homeostatic and regeneration data is combined instead of analyzed separately (before and during regeneration) to compare changes in (1) the hair cell populations and (2) the transcriptomes of these hair cell populations. Either homeostatic and regeneration conditions should be analyzed separately (hence, 2 different figures), or the authors should explain why the data sets were combined for this analysis. Figure 2a and 2b.

“Clusters 4 (yHC) and 5 (mHC) encompassed the hair cell (HC) lineage and expressed the *atoh1a*, *s100t*, and *pvalb8* genes (Figure 2F).” From the UMAP in figure 2A there are two populations of HC that are not described, HC4 and HC5.

Figure 2B heat map shows the genes expressed in HC4 and HC5. Should these two populations be one cluster based on the heat map in figure 2B and why are they not described in the text? Also, HC4 and HC5 are not described until Figure 2F where in the text they are defined as yHC and mHC but are not on the UMAP (labels should be consistent in the text and figure).

What genes define the separation of these two clusters to make one young and mature hair cells? The heat map and figure 2F are not clear to show this distinction.

“For each cell population we identified genes specifically expressed or highly enriched (Figure 2B). We controlled for global gene expression responses caused by dissection, dissociation and other manipulations by selecting genes that were variable between clusters and/or between samples.” How was this done – in brief – what package – and what was used as a cut off to determine which genes were highly expressed?

“Based on marker gene expression, we detected 2 populations of support cells (clusters SC0 and SC2) which showed significant enrichment for the markers *cldn7b*, *epcam*, *lima1a*, and *hbegfa* (Figure 2D).” Actually from figure it looks like *epcam* and *cldn7b* and more not limited at all to just the SC0 and SC2. – please clarify

“The zebrafish auditory (sacculle) and vestibular (utricle) organs are distinct sub-organs that share morphological and physiological aspects. The utricle in fish primarily functions as a gravitation sensor but has also shown auditory potential (Popper, 2011; Schulz-Mirbach and Ladich, 2016).” Too late here! Discuss in the intro.

The gene *otofa* for example has higher expression in the utricle compared to the sacculle, yet both tissues express this gene during hair cell differentiation (Figure S2). Since the regeneration programs between organs were so similar, we decided to combine sacculle and utricle data for subsequent analysis to increase cell sample size and boost statistical power. The logic here is lost on me. There’s a difference but they are the same so data consolidated. Can this be explained better?

“Although Uniform Manifold Approximation and Projection’s (UMAPs) of individual cell types show that the cells in non-regenerating and regenerating tissues cluster together (Figure 3A).” You mention regenerating tissue. What time point of regeneration is this? Are these all time-points combined or a single time point during regeneration? Needs to be clearer.

It is also not clear why the number of cells during regeneration increases past the number of cells at baseline (non-regeneration).

the regenerating inner ear exhibits an expansion of a class of cells that were intermediate between support cells and hair cells (“PC2” in Figure 3B) Could be true but hard to see in the figure. Can this be magnified or meaningfully quantitated? This is not clear. PC2 is in non-regeneration. During regeneration, all clusters seem to expand in number of cells. This is not just unique to PC2. The way this is written, it sounds like this expansion of cells is only unique to PC2?

“Hair cell specific genes such as *s100t* and *pax2a* were upregulated in supporting cells and in young or mature hair cells. Young and maturing hair cells exhibited an upregulation of *myo6b*, *s100s*, and *pou4f1*.” How is significance calculated?

Figure 4 – confusing about numbers of clusters – can the numbers be consistent?

Figure 4- should have a simple diagram of the Pseudotime. Perhaps a model of some sort.

Figure 4 legend “The point where support cells transition to progenitor cells is labeled “PC” There is no clear labeling of PC as the authors state these cells are labeled in the figure.

We found that the aggregate of scATAC-seq profiles closely resemble bulk ATAC-seq samples, indicating that aggregate scATAC-seq captured the chromatin accessibility in a manner equivalent to bulk ATAC-seq assays (Figure 5A). Global comment but figure is specific example – could this be addressed in text better.

A hallmark of high quality ATAC-seq libraries is a banded insert size distribution with peaks or genomic signals representing putative regulatory regions resulting from nucleosome protection, which was apparent even in individual cells (Figure 5B). Reference?

Indeed, we found a clear correlation between chromatin accessibility and nearby/distal gene expression (p value < 0.05 , hypergeometric Test) (Figure 6D and Figure 6E; Table S15). Here and in other parts of this paper – direct numbers are used rather than proportions/percentages of cells. 6D is the number of DE genes that are also linked to RREs. Don't we need to see the percentage of DE genes that also intersect with a RRE? Just a number seems impossible to interpret unless you look back to the number of genes mentioned pages ago.

Enhancer deletion mutants had no overt morphological phenotypes in early larvae, and both heterozygous and homozygous deletions survived to adulthood. To interpret this section better, we need to know how much *Sox2* is normally expressed during development and adult homeostasis. Is the enhancer

regeneration specific or is still used at other times but function not critical?

Figure B shows normal larvae have less hair cells to begin with – although no stats shown for that comparison.

In non-regenerating sensory epithelia of the enhancer deletion mutants, *sox2* RNA levels were not significantly altered in comparison to control sensory epithelia. In adult zebrafish without the enhancer deletion undergoing hair cell regeneration, *sox2* expression was elevated on days 4, 5, and 7 post-DT. By day 9 post-DT, *sox2* expression levels were near control levels and were further reduced on day 11 post-DT (Figure 9D). What?

In non-regenerating sensory epithelia of the enhancer deletion mutants, *sox2* RNA levels were not significantly altered in comparison to control sensory epithelia. Where does this result come from? In adult zebrafish without the enhancer deletion undergoing hair cell regeneration, *sox2* expression was elevated on days 4, 5, and 7 post-DT. Figure D? Figure 9E – presumably a model but no explanation or legend. No idea what the spots represent.

Materials and methods- “aligned to the zebrafish genome” Which zebrafish genome was used to align the data to?

Authors' response to the first round of review

We would like to thank all the reviewers for both their overall support of our work and the high quality of the comments that have resulted in a much stronger manuscript. Our specific responses to each reviewer are detailed below.

*Reviewer #1: This manuscript by Jimenez et al. is an elegant approach to identify gene expression and cis-regulatory changes accompanying regeneration of mechanosensory hair cells in the zebrafish inner ear. Single-cell transcriptome and chromatin accessibility are assayed during hair cell regeneration following genetic ablation in adults. Strengths of the study include looking directly at the inner ear rather than the more commonly studied lateral line that has less homology to the mammalian inner ear. A large number of regeneration-specific enhancers are also identified, and interestingly many of these show sequence conservation across fish species, suggestive of an ancient regulatory program for hair cell regeneration. The most striking aspect of the study is the identification of a regeneration-specific enhancer for *sox2*, which when deleted affects the timing of *sox2* expression and*

long-term hair cell regeneration. This is quite a surprising result given that knockout of regeneration-specific enhancers by other groups have not shown regeneration defects, that heterozygous sox2 enhancer deletion has the same regeneration effect as homozygous deletion, and that enhancer deletion results in only subtle shifts in the timing of sox2 expression. However, characterization of this enhancer deletion, including possible off-target effects, is not adequately discussed. If further data can be generated to support specificity of this enhancer deletion, including analysis of independent alleles, then this study will be of considerable interest to not only the hearing research community but the stem cell and genomics communities overall.

- 1. The exact sequence of the sox2 enhancer deletion should be shown. It is a bit surprising that heterozygotes show the same defect as homozygotes. Were more than one independent deletion allele analyzed? CRISPR is known to cause large inversions and other chromosomal abnormalities that may be difficult to detect by PCR, as well as off-target lesions. The result should therefore be validated with at least two independent alleles, extensive outcrossing should be performed to show the phenotype segregates with the intended deletion, and Southern blot should be used to rule out local chromosomal rearrangements (though similar phenotype in independent alleles would make this unnecessary).*

Author Response: We agree with Reviewer #1's comment that it is surprising that the sox2 enhancer deletion is haploinsufficient. We also agree that the characterization of this enhancer deletion, including possible off-target effects can be addressed with additional alleles. We generated 3 independent sox2 enhancer deletion mutants: sox2hg138, sox2hg139, and sox2hg140. To identify the molecular lesion of the CRISPR-induced deletion, PCR products were amplified from DNA of F1 zebrafish containing heterozygous deletions from the three independent alleles. PCR products were purified and subcloned into pCR2.1-TOPO vector by TA cloning. Following bacterial transformation, 4 colonies were picked and grown overnight for each allele. Plasmid DNA was extracted and analyzed by Sanger sequencing. The sequence of each deletion allele is shown in Figure S7A. We also show the deleted region in Figure 6C and Figure S5. The sequence coordinates of the deleted region are described in the legend of Figure S5. We Response to Reviewers confirmed the lateral line hair cell regeneration phenotypes for each of the independent alleles. We

demonstrated that progeny from heterozygotic F2 sox2hg138, sox2hg139, and sox2hg140 in-crosses exhibited similar phenotypes in both heterozygous and homozygous backgrounds (Figure S7B). We have revised the results, methods, and discussion sections accordingly.

2. *In Fig 9C, D and Fig S6, it is unclear whether hets and/or homos are being examined, in contrast to the text which states both are being examined.*

Author Response: We have revised the text to clarify what is being tested and have added descriptions of the genotypes in the figure legend and text.

3. *In the Discussion, a definitive model is put forth where SCs become PCs that enter mitosis, and then either become HCs or revert back to SCs. This can only be inferred from pseudotime analysis alone and should be greatly softened. Only definitive in vivo lineage tracing (and proliferation assays) can be used to confirm these bioinformatic predictions.*

Author Response: We agree with the reviewer that the transition from SC to PCs can only be inferred from pseudotime analysis alone. Therefore, we have softened the text as suggested on pg. 14 line 42: "From this analysis it is unclear if the progenitor cells emerge from the support cells and can return to support cell identity or if there are resident progenitor cells that expand and either become hair cell or support cells." And in the discussion we state (pg. 14, line 36): "From this analysis it is unclear if the progenitor cells emerge from the support cells and can return to support cell identity or if there are resident progenitor cells that rapidly expand after an injury and either become hair cells or support cells. Even in uninjured inner ears, we detected progenitor cells, we can envision two reasons why this may be, 1) it has been shown in adult zebrafish that hair cells continually grow throughout the lifespan, the progenitor cells could be always present because they are in the process of transitioning from support cells to hair cells during that growth, or 2) the progenitor cells in fish represent a cache of multipotent stem cells that contribute to the constant growth and can also replenish support cells and/or hair cells after injury or death. Understanding how support cells give rise to new hair cells in the adult inner ear of zebrafish may contribute to a better understanding of overcoming the regeneration block in the mammalian inner ear."

4. *In the Abstract, the term "pluripotent" should be reserved for cells that can generate all cell types of the embryo. PCs in the inner ear are multipotent,*

i.e. generating hair and support cells only. Along these lines, the comment that "Sox genes in general are associated with pluripotent cells" is irrelevant and should be deleted.

Author Response: Reviewer #1 is correct and we have deleted the comment as suggested and revised the text accordingly.

- 5. Differences between the adult inner ear and larval lateral line could either be due to the different stages (adult/larvae) or the type of mechanosensory organ. Conclusions should be toned down in the absence of stage-matched comparisons. Also analysis in Fig S1 is rather cursory.*

Author Response: We agree that the analysis in Fig S1 is brief and while it seems obvious that adult zebrafish inner ear is transcriptionally distinct likely due to the type of mechanosensory organ, most of the research to elucidate the mechanisms involved in hair cell regeneration presumes that the responses in the larval lateral line can be applied to adult inner ear sensory epithelium *in vivo*. While the larval zebrafish lateral line might be a great proxy to screen for hair cell regeneration factors and understand lateral line hair cell regeneration, our initial observations described in the text related to Fig S1 point to a need to investigate hair cell regeneration in the context of the adult inner ear organ. Considering that the adult zebrafish inner ear shares common features to a mammalian inner ear like auditory and vestibular organs, the adult inner ear transcriptomes of zebrafish are likely more comparable than a lateral line and our data will be useful for comparative studies between zebrafish and mouse. We also think the comparison between adult inner ears and larval lateral line could be valuable in identifying “core” aspects of hair cell regeneration even though support and precursor cells likely possess significant differences in ground state identities. Nevertheless, we have toned down our conclusions as advised. Pg. 5 line 24: “The differences in transcriptome support the transcriptional distinctiveness of adult inner ear sensory epithelia from their larval lateral line counterparts due to differences in mechanosensory organ or age of stages examined suggesting a value to examining the core conserved pathways between the two organs as well as the ones unique to each organ.”

- 6. The switching on of sox2 expression during regeneration in Fig S4 is quite subtle. Other sox genes also appear to switch on in a similar manner. Why is there no genetic redundancy of sox factors? This should be discussed.*

Author Response: It is true that further elaborating on the genetic redundancy or lack thereof of sox factors may be insightful and have now presented this in the discussion section. Pg. 15, line 39: "For both the Sox and Six transcription factor families, we see more than one being expressed at any given timepoint during regeneration. It may be that this represents some functional redundancy within the family members. It will be informative to test this possibility in the future by not only performing single gene deletions, but also in various combinations to see if the effects of deleting two or more transcription factors are more than additive. In at least with the case of Sox2 function, simply delaying expression by 24 hours is sufficient to severely inhibit regeneration, suggesting that if there is some redundancy with other Sox transcription factors, it isn't nearly sufficient to compensate for the loss of Sox2 function."

7. *Fig 8. For the majority of genes shown, upregulation during regeneration is not apparent. It would also be helpful to annotate cluster identity.*

Author response: We have modified Figure 8 to include a panel that accompanies cluster identification with specific genes within each cluster demonstrating the gene expression dynamics. We observe an upregulation in expression of the six genes (particularly for six1a and six1b), while the sox genes are more dynamic in expression. Some sox genes (sox2, sox21) turn on in progenitor cells and turn off in hair cells during regeneration. While the gene expression during state switching can be most appreciated in the pseudotime analysis we describe in the manuscript, we have included violin plots to show the expression of representative genes observed in non-regenerating vs. regenerating conditions.

Minor comments:

1. Define "S" and "U" abbreviations in Fig 2 legend.

Author Response: We have revised the Figure 2 legend with definitions for "S" and "U".

2. Hard to appreciate expression score of Module 3 in mature HCs in Fig 4D.

Author Response: Module 3 shows trajectory variable genes with one-dimensional coordinates that reflect its state in the regeneration process. Module 3 trajectory variable genes have a significant negative

correlation with pseudotime coordinates and this represents genes that essentially only turn on in the “end state” of what we predict is a mature, functional hair cell. The genes in this module in Table S8 are genes like *atoh1a* and *six1b* which are actually turning off in mature hair cells. There is a clear difference between Module 3 and “quiet” modules like Modules 6. To help the reader appreciate the data presented in Figure 4, we have revised the text and added the line on pg. 8 line 4: “Genes grouped in module 3 included many negatively regulated genes such as *six1b* and *atoh1a* which are known to be downregulated in mature hair cells.”

3. The method used to integrate scATAC and scRNA should be explained in main text as this was not a multiome experiment. Discussion of this integration in Methods is insufficient. Was an algorithm such as SNAP-ATAC used?

Author Response: We have revised the main text to clarify that samples were processed as separate assays for scRNA-seq and scATAC-seq experiments in different cells. We have also revised the methods section so that it is appropriate for STAR Methods. In the methods section, we describe the scRNA-seq and scATAC-seq methodology. In response to Reviewer #1 regarding SNAP-ATAC: The two modalities (scRNA-seq and scATAC-seq) were integrated using Seurat and Signac methods for cross-modality integration and label transfer and we included a separate section in the STAR Methods section that includes a description of the integration via Seurat and Signac.

4. It would be helpful to annotate the clusters in Fig 5D. Also the dot plots in Fig 5E-G are not easy to interpret - many are very low expression and do not appear localized to specific regions as stated in text.

Author Response: We annotated the clusters in Figure 5D as suggested. In regards to Figure 5EG being difficult to interpret: We agree with Reviewer #1 that the dot plots in Figure 5E-G are difficult to interpret and recognize this limitation of visualizing the scATAC-seq measurements (peaks). Visualizing activities of marker genes to help interpret scATAC-seq clusters is noisier than scRNA-seq measurements because the activities from scATAC-seq represents measurements from sparse chromatin data due to the intrinsic limitation of numbers of chromosomes per nucleus and because they assume a general

correspondence between gene body/promoter accessibility and gene expression which may not always be the case (references: <https://pubmed.ncbi.nlm.nih.gov/26294014/>). We recognize this limitation should be mentioned in the paper, so we added the following sentence (pg. 9 line 7): “Visualizing activities of marker genes to help interpret ATAC-seq clusters was noisier than scRNA-seq measurements because the activities from scATAC-seq represents measurements from sparse chromatin data.”

5. ZF/GF/C and CNE should be defined in legend to Fig 6.

Author Response: We have revised the Figure 6 legend with the appropriate information.

6. The discussion of machine learning to find enriched motifs in enhancers is misleading, as from the Methods it appears that this is just the Homer algorithm.

Author Response: We have clarified the Methods section because the Machine Learning used was not the HOMER algorithm. HOMER is a suite of tools that is not only used for Motif discovery but can also be used for other forms of sequence analysis. HOMER provides a utility for comparing sets of ATAC-seq peaks called mergePeaks. We used the HOMER mergePeaks function to identify overlapping peaks and to merge peaks together into a single file to use for downstream analysis, one of which was machine learning to identify TF motifs. These merged peaks were also fed into the GREAT analysis. We have revised the methods section to clarify how each program was used.

7. Fig 7A SC, it is confusing why a lack of link between SOX10/SOX9 and SOX10/SOX9 is shown, whereas links between SOX and SIX are show for PC and HC.

Author Response: On the horizontal and vertical axes are the names of all the transcription factor binding sites significantly enriched in each cell type. You will notice that the horizontal and vertical axes are different among the three plots because there are different transcription factor binding sites enriched in a cell type specific manner, so the TF motifs and their order are not identical in each plot. If the transcription factor binding site on the horizontal and vertical axes intersect AND they also co-occupy a regulatory region or can be found

within the same regulatory sequence, then the box is colored. The points of intersection between the same transcription factor (on the diagonal axis) are not colored in because it isn't a co-occupancy by definition. SOX10/SOX9 is a predictive category that doesn't differentiate between the two genes as their binding sites are similar enough that they cannot be distinguished computationally. There may be some interactions between Sox9 and Sox10, but they cannot be determined using this method. To aid the reader, we walked through the interpretation of the plots in the Figure 7 legend (pg. 19, line 7: "Pair-wise predicted TF co-association in support cells, progenitor cells, and hair cells. Horizontal and vertical axes show transcription factor categories of significantly enriched predicted TF binding sites. Grey boxes represent lack of co-association between the TFs denoted in horizontal and vertical axes whereas the colored boxes represent enrichment of co-association between the predicted TF sites compared to chance. Intersections of the same TF category are colored grey because they cannot, by definition, interact.")

8. Fig 9. Panel E model is not described in legend. The dots panels are therefore hard to interpret.

Author Response: We apologize for the error in the initial submission. We have now described the panel model in the legend of Figure 9. We have also modified the schematic diagram to aid the reader. In the revised submission, the model is now in Panel F and the text is revised accordingly.

9. The sox2 deleted element should be indicated in Fig 6C.

Author Response: We have added the sox2 enhancer deletion element in Figure 6C and revised the text and legend.

10. There appear to be 3 CNE in Fig S5, not only 2 as described in text.

Author Response: Reviewer #1 is correct, and we have modified the text and legend. _____

Reviewer #2: In this manuscript Jimenez et al present a "tour de force" of single cell and ATACseq data on isolated cells from the adult inner ear cells and larval lateral line cells from the zebrafish. This is the first time such a careful comprehensive comparative analysis has been carried out and there is an amazing

amount of new data presented in this manuscript which opens up many new avenues of research. The authors have done a commendable job of using AI to extract really meaningful biologically relevant data from these huge datasets. From this they have then zoned in and identified a sox2 enhancer element which appears to be specific to hair cells regeneration, they present elegant CRISPR knock-out data showing how this mainly conserved enhancer is essential for normal hair cell regeneration. In theory the paper could be published as is as it is very suited to a journal focusing on Genomics. A few minor revisions would broaden the impact of the work:

1. Figure 1 and 9 need "n = x" for all biological experiments added to the figure legend. Author Response: We have added this important piece of information in the figure legends of Figure 1 and 9. 2. The authors refer in the manuscript "differentiated Sox2 positive support cells are left undamaged". Sox2 is a marker of neural stem cells, so it seems the work "differentiated" is not appropriate in this context and should be removed.

Authors Response: We have made the requested changes in the text to avoid confusion by readers, but there is expression of sox2 in the differentiated zebrafish inner ear support cells (figure 1) and it has been documented in adult mouse support cells in the cochlea as well (Hume et al., 2008

<https://www.sciencedirect.com/science/article/pii/S1567133X07000476?via%3Dihub>).

2. Figure 9 shows interesting data to suggest that when the sox2 enhancer is deleted this results in a delay in the regeneration process, is this due to a delay in initiation of the regeneration response due to a delay in proliferation of the support cells in response to injury?

Author Response: This is an important question that we would like to address in our next publication characterizing the sox2 enhancer deletion mutant and the molecular mechanism of enhancer function on hair cell regeneration. Reviewer #2 raises an important question that we started to address in the lateral line system of enhancer deletion mutants (data not shown or described in this paper). Preliminarily, we used the FUCCI transgenic system which differentially marks cells in G1 and S/G2/M phases of the cell cycle and can therefore be used to separate rapidly and slowly cycling cells in vivo by the virtue of the cell cycle state they primarily

inhabit. Although the system cannot be used to distinguish the cells in the G0 phase from those in the G1, we have examined cells in the G1-S and M-G1 transitions and found that there is not a significant difference in the average number of G1-S cells per neuromast in regenerating (24 hrs post CuSO₄) *sox2* enhancer deletion mutants. Experiments are underway to assess mitotic events in neuromasts of the lateral line during the first 24 hours of the regeneration period using BrdU incorporation assays.

3. Does repeated hair cell ablation over a period of weeks/months in the CRISPR *sox2* enhancer knock-out animals led to a more striking regeneration defect, is the enhancer element needed to maintain and replenish the support cells?

Author response: This is an intriguing question that only has been addressed by experimentation in the lateral line hair cell system of wild-type adult zebrafish. An assumption based on the lateral line is that the adult zebrafish inner ear hair cells also undergo continuous turnover in the absence of damage (it has been shown that the number of hair cells increases in zebrafish over the entire lifespan of the animal). An important object to study is to determine if the adult zebrafish inner ear can maintain hair cells and support cells after multiple rounds of damage and regeneration. It will be critical to address if the *sox2* enhancer element is needed to maintain and replenish the support cells and we would like to address this in a future publication. Based on our observations on the enhancer deletion mutants 23 days after just a single treatment of DT, *sox2* positive support cells are not significantly compromised compared to controls. Considering that the enhancer appears to regulate the timing of *sox2* gene activation, we predict that repeated hair cell ablation over a period of weeks/months in the CRISPR *sox2* enhancer mutant will lead to a striking defect. There are many aspects that we would like to address in a separate manuscript describing the molecular mechanism of *sox2* enhancer regulation on hair cell regeneration including this important question Reviewer #2 has raised.

Reviewer #3: A regulatory network of Sox and Six transcription factors initiate a cell fate transformation during hearing regeneration in adult zebrafish The research described in this manuscript is a tour de force and is worthy of publication in this journal. However, the actual writing is something of a surprise. I would suggest that the authors take another look and make some adjustments

in particular the short introduction needs to explain zebrafish hair cell biology better to make the paper more accessible to a genomic audience. For such a magnitude of results, the discussion is also underwhelming. Everything mentioned below is a minor point – but this was not an exhaustive list.

Results: “L injection of diphtheria toxin (DT) leads to complete hair cell loss of auditory (sacculle) and vestibular (utricle) hair cells 5 days-post injection leaving a population of differentiated sox2 positive supporting cells undamaged (Jimenez et al., 2021) (Figure 1A).” Looking at both the figure and their other paper this is an unnecessary over statement. You can see a few green cells at every stage. For this work 100% ablation was not required – reword.

Author Response: We agree with Reviewer #3 and have revised the text accordingly. Figure 1 A diagram of what we are looking at would make it easier for those not familiar with zebrafish ear anatomy. Figure 1b There are no stats or “n” described for the experiment done. Why SEM not SD? Author Response: In response to Reviewer #3’s comments regarding Figure 1, a diagram (now Figure 1A) of the adult zebrafish inner ear was added to help explain the structure of the inner ear described in the text. To complement the graph and quantification of phalloidin positive hair cells in Figure 1C, two-way ANOVA statistics was performed, error bars show Standard Deviation, and “n” are now described in the figure legends.

“We dissected out sacculles and utricles at three time points: Days 4, 5 and 7 post-DT, and each were processed separately in both scRNA-seq and scATAC-seq experiments.” This anatomy has yet to be described or significance mentioned (although it is addressed better later in the text- add to intro

Author Response: We have revised the introduction and added a description of the anatomy and discussed the significance. Pg. 2 line 43: “In fish these organs are the sacculle and utricle, respectively (Figure 1A). The sacculle and utricle are distinct sub-organs that share morphological and physiological aspects of hearing and balance similar to their mammalian counterparts. The sacculle in fish primarily detects acoustic vibrations (amplified through the body instead of via an eardrum) while the utricle primarily functions as a gravitation sensor but has also shown some auditory potential” Figure 1c - repeats some data from 1a and adds

info for utricle. However, the resolution is now much lower and doesn't really add much. Perhaps 1c should just be the new info for utricle and at higher res.

Author Response: In response to Reviewer #3 regarding what was Figure 1C (and now Figure 1D), we would like to justify including the panel in Figure 1 for publication. Although the resolution is much lower, the figure panel displays properties of the samples used and aims to provide a visual of the entire structure (top to bottom) of the organs, the saccule and utricle. To help the reader, we have added diagram drawings of the saccule (red) and utricle (blue) next to panel 1D to provide a visual display of the tissues harvested for genomics and functional analysis experiments.

Major plus that two negative controls are used in ablation – controlling for both DT and the transgene (presumably the expression of a human receptor) - it would be great if the premise were explained better and more info given when each control is used and why.

Author Response: We have revised the text to discuss our rational for using the two negative controls and more information was provided for why each control was used. Pg. 4 line 8; “To control for the presence of the hDTR transgene and diphtheria toxin (DT) treatment, untreated Tg(myo6b:hDTR) transgenic zebrafish and wild-type DT injected (Day 4) zebrafish lacking the Tg(myo6b:hDTR) transgene were used as homeostatic or non-regenerating controls. Untreated wild-type zebrafish were also used as a non-regenerating control.”

“We generated twelve transcriptomic profiles on inner ear tissue using the 10X Chromium system for droplet-based single-cell 3' RNA sequencing (scRNA-seq) and quantified each data set using the Cell Ranger 6.0.0 pipeline (10x Genomics).”
What is meant by 12 transcriptomic profiles? 12 replicate experiments? 12 different conditions?

Author Response: We have revised the text to explain that 12 transcriptomic profiles correspond to the 12 sample conditions described.

“Having established the different support, progenitor, and hair cell populations of the inner ear allowed us to interrogate the dynamic expression pattern of specific genes of interest in nonregenerating and regenerating inner ear tissues.” It is stated that you have established these conditions in non-regenerating and

regenerating inner ear tissues. This was not done very clearly. The data presented in figure 2 combines homeostatic scRNAseq data with regeneration data. It is not clear why homeostatic and regeneration data is combined instead of analyzed separately (before and during regeneration) to compare changes in (1) the hair cell populations and (2) the transcriptomes of these hair cell populations. Either homeostatic and regeneration conditions should be analyzed separately (hence, 2 different figures), or the authors should explain why the data sets were combined for this analysis.

Author Response: We have revised the text for clarity. The purpose of Figure 2 is to show that the cells from the 12 independent scRNA-seq samples cluster together into an atlas that represents the adult zebrafish inner ear, i.e. during regeneration we don't see the emergence of new cell populations, only changes in the distributions of the different cell groupings. Unsupervised cell clustering is the most common approach in single cell genomics analysis and one goal is to first identify the cell types present in a sample among other quality control measures before proceeding to differential testing. Identifying cell populations that are present across multiple datasets can be problematic and challenging. But by integrating all of the scRNA-seq samples into one aggregate composed of all 12 different samples, we constructed a harmonized atlas of the adult inner ear. We have revised the text to explain our analysis pipeline more clearly. Pg 4, line 23: "After filtering, we integrated scRNA-seq datasets from all twelve separate samples for cell type clustering in order to identify cell types based on conserved gene expression and to demonstrate that all twelve samples cluster together into a harmonized atlas of the adult zebrafish inner ear sensory epithelia. Samples were assembled into an aggregate or "unified transcriptomic atlas" for clustering. Unsupervised clustering partitioned 66,296 inner ear sensory epithelial cells (sacculle and utricle combined) into discrete scRNA-seq cell populations (Figure 2A; Table S6). We assigned cell type identities to each of these clusters based on the top differentially expressed genes associated with each cluster and the known expression of these genes based on the literature using zebrafish transcriptome data from inner ear cells 26 and single-cell transcriptome data from the larval lateral line." To compare changes between homeostatic or non-regenerating versus regenerating conditions, we performed pair wise comparisons between homeostatic or non-regenerating controls and regenerating sensory epithelia at

each time point within the clusters (Figure 3). We have revised the main text and methods section to clarify in response to Reviewer #3's comments. Figure 2a and 2b. "Clusters 4 (yHC) and 5 (mHC) encompassed the hair cell (HC) lineage and expressed the *atoh1a*, *s100t*, and *pvalb8* genes (Figure 2F)."

From the UMAP in figure 2A there are two populations of HC that are not described, HC4 and HC5. Figure 2B heat map shows the genes expressed in HC4 and HC5. Should these two populations be one cluster based on the heat map in figure 2B and why are they not described in the text? Also, HC4 and HC5 are not described until Figure 2F where in the text they are defined as yHC and mHC but are not on the UMAP (labels should be consistent in the text and figure). What genes define the separation of these two clusters to make one young and mature hair cells? The heat map and figure 2F are not clear to show this distinction.

Author Response: We agree that the explanation is unclear. In response to comments regarding the heatmap, we used Seurat to perform clustering and to generate the single cell heatmap of feature gene expression. Clusters HC4 and HC5 could potentially be combined to be one cluster based on the heatmap. It is important to note that for scRNA-seq analysis with Seurat, Seurat clusters cells based on their PCA (Principle Component Analysis) scores, with each PC representing a metagene that combines information across a correlated gene set. Determining how many PCs to include downstream is an important step and was done for this analysis based on plotting the standard deviation of each PC (https://satijalab.org/seurat/archive/v3.0/pbmc3k_tutorial.html). Looking at the heatmap in Figure 1B, we agree that instead of 11, having 10 clusters might seem more appropriate. However, Seurat applied a graph-based clustering approach embedding the cells in a graph structure and "drew" edges between the cells with similar feature expression patterns. Seurat generated the cluster 4 (HC4) and cluster 5 (HC5). The differences between cluster 4 (HC4) and cluster 5 (HC5) may seem arbitrary, but if we look closer at what makes these cell types different, we find that cluster 4 (HC4) expresses markers such as *rpl* and GO analysis enrichment for translation and ribosome assembly. While cluster 5 (HC5) does not express these marker genes. Since synthesis of ribosomal proteins is shut down in mature hair cells of the neuromast in the lateral line, cluster 5 (HC5) was identified as "mature" hair cells (Lush et al., 2019). In response to Reviewer #3's comment, we have revised the text explaining the differences between cluster 4

(HC4) and cluster 5 (HC5). Pg. 5 line 4: “Cluster 4 (HC4) was identified as young hair cells because this cluster showed significant enrichment for the markers *rpl* and GO analysis enrichment for translation, ribosome assembly while cluster 5 (HC5) did not. Since synthesis of ribosomal proteins is shut down in mature hair cells of the neuromast in the lateral line, cluster 5 (HC5) was identified as mature hair cells.” In addition, we have revised all labels in inconsistent figures. The labels are now consistent throughout the text, figure legends, and figures. By reviewer #3 bringing up this point it also makes us realize we are seeing a very large number of cells classified as hair cells in the regenerating tissue, many more than would make sense in terms of the final number of hair cells we see when the regenerating tissue reaches homeostasis again. We now mention this in the discussion (pg. 15, line 1): “Another interesting observation is that we are seeing many more cells in the hair cell clusters than could realistically become hair cells when the tissue returns to homeostasis. It will be important to determine if these excess hair cells are actively “pruned” from the sensory epithelium once the correct number of hair cells are established or if immature hair cells go through a process of “dedifferentiating” back to precursor cells or even to supporting cells.”

“For each cell population we identified genes specifically expressed or highly enriched (Figure 2B). We controlled for global gene expression responses caused by dissection, dissociation and other manipulations by selecting genes that were variable between clusters and/or between samples.” How was this done – in brief – what package – and what was used as a cut off to determine which genes were highly expressed?

Author Response: We revised the main text and methods section to include the line in the methods: Highly variable features were identified using the `FindVariableFeatures` function (`selection.method = “vst”`) followed by linear transformation using the `ScaleData` function on all genes. We revised the main text and have included the following line: Using the `FindVariableFeatures` function in the Seurat pre-processing and integration procedure (27), we controlled for global gene expression responses by selecting genes that were variable between clusters (i.e, they were highly expressed in some cells and lowly expressed in others). “Based on marker gene expression, we detected 2 populations of support cells (clusters SC0 and SC2) which showed significant enrichment for the markers *cldn7b*, *epcam*, *lima1a*, and *hbegfa* (Figure 2D).”

Actually from figure it looks like epcam and cldn7b and more not limited at all to just the SC0 and SC2. – please clarify

Authors Response: We agree with Reviewer #3 that the support cell markers chosen (epcam, cldn7b, hbegf, and lima1a) do not represent markers specific to the support cell population, although they are highly represented. While expression of epcam, cldn7b, hbegf, and lima1a are enriched in the support cells and collectively can define the population, these genes are also can be expressed in PCs, although significantly weaker in expression. We therefore revised Figure 2D and present support cell markers that are among the top 10 highly expressed genes in either clusters 0 (SC0) or cluster 2 (SC2). We have also revised the text and included that differential expression (DE) testing for cell type markers was accomplished using Seurat (FindMarkers or FindAllMarkers; Wilcoxon rank sum test).

“The zebrafish auditory (sacculle) and vestibular (utricle) organs are distinct sub-organs that share morphological and physiological aspects. The utricle in fish primarily functions as a gravitation sensor but has also shown auditory potential (Popper, 2011; Schulz-Mirbach and Ladich, 2016).” Too late here! Discuss in the intro.

Author Response: As mentioned above, we have revised the introduction. The introduction now includes background about inner ear sensory epithelia and its significance. “The gene otoa for example has higher expression in the utricle compared to the sacculle, yet both tissues express this gene during hair cell differentiation (Figure S2).

Since the regeneration programs between organs were so similar, we decided to combine sacculle and utricle data for subsequent analysis to increase cell sample size and boost statistical power.” The logic here is lost on me. There’s a difference but they are the same so data consolidated.

Author Response: We apologize for the original description of how we evaluated the data in the initial submission. We have carefully evaluated the expression of the genes mentioned in Supplemental Table S3 and 5 and have revised the text (pg.5 line 1): “We performed differential (DE) expression testing for cell types between sacculle and utricle datasets using Seurat and the Wilcoxon rank sum test. In agreement with Yao et al., we observed elevated expression levels of

wnt11 (formerly wnt11r 30), sema3e, otol1a, and nr2f1, and vwa2 globally in homeostatic or nonregenerating saccule but these genes are not differentially expressed in support cell, progenitor, or hair cell clusters specifically in regenerating sensory epithelia (Table S5). Similarly, the gene otofb for example has elevated expression in the hair cells (HC8) of homeostatic or non-regenerating utricle compared to the saccule (p-value < 0.02; FC = 0.34) (Table S3), yet the regeneration responses in the hair cells are comparable and no statistically significant difference in otofb expression is observed (Table S5). Since the regeneration programs between organs were so similar, we decided to combine saccule and utricle data for subsequent analysis to increase cell sample size and boost statistical power.” “Although Uniform Manifold Approximation and Projection’s (UMAPs) of individual cell types show that the cells in non-regenerating and regenerating tissues cluster together (Figure 3A).”

You mention regenerating tissue. What time point of regeneration is this? Are these all timepoints combined or a single time point during regeneration? Needs to be clearer. It is also not clear why the number of cells during regeneration increases past the number of cells at baseline (non-regeneration). Could be true but hard to see in the figure. Can this be magnified or meaningfully quantitated? This is not clear. PC2 is in non-regeneration. During regeneration, all clusters seem to expand in number of cells. This is not just unique to PC2. The way this is written, it sounds like this expansion of cells is only unique to PC2?

Author Response: In response to Reviewer #3, we have explained more in the text why all time points were combined for the data in Figure 3. In the revised text and Figure 3, we combined all control samples (uninjected and DT injected WT utricle and saccule; uninjected DTR transgenic fish utricle and saccule) and compared them to regeneration samples (Days 4, 5, 6 post-DT injected DTR transgenic fish utricle and saccule). Using Seurat and R, we also quantified the number of cells in the non-regenerating aggregate and the regenerating aggregates in Figure 3. Although it appears that the number of cells during regeneration increases past the number of cells at baseline (non-regeneration), it’s not the case. For the analysis in Figure 3, there are 3,670 cells total in the non-regenerating control aggregate and 3,330 cells total in the regenerating aggregate. We also meaningfully quantitated the number of cells in each cluster of nonregenerating and regenerating groups. In agreement with Reviewer #3, cell expansion is not

exclusive to progenitor cells. We find that there is an expansion of progenitor cell and hair cells. We revised the text and included the line (Pg. 6 line 27): “In comparison to non-regenerating tissue, the regenerating inner ear exhibits a decline in support cells (“SC0”), an increase of a class of cells that were intermediate between support cells and hair cells (“PC1” in Figure 3B), and an increase of hair cells (“HC2”) (Figure 3B).”

“Hair cell specific genes such as *s100t* and *pax2a* were upregulated in supporting cells and in young or mature hair cells. Young and maturing hair cells exhibited an upregulation of *myo6b*, *s100s*, and *pou4f1*.” How is significance calculated?

Author Response: We have revised the text and described how significance was calculated using the Seurat differential expression test algorithm. We have included an explanation in the methods section and have added the p-values and fold changes produced by the Seurat statistical testing algorithm in the main text, methods, and figure legends.

Figure 4 – confusing about numbers of clusters – can the numbers be consistent? And also the weird ordering in 1A I would suggest to them that the clusters should be labeled as they did in the above figures

Author Response: Regarding the comment for 1A, we have revised the labels so that they are consistent throughout the figures and text. Regarding Reviewer #3’s comment, “can the numbers be consistent?” Although strict hierarchical clustering is not used in the single cell analysis workflow applied in this work, the Monocle 3 algorithm used does have tunable parameters whose user defined adjustment can produce very different results and it can be hard to address the quality of the chosen clusters or whether the cells have been over or under clustered. If there were less clusters, its possible that there would be an under-clustering where the clusters are too broad and mask underlying biological structure. If the cells are over clustered, non-relevant subdivisions can be introduced, but the subclusters can be merged to recover appropriate cell types. The right resolution for clustering depends on the data and needs during downstream analysis. In our case, we think that we have reached near optimal clustering with Monocle 3 using a resolution of $1e-2$ because there are relevant biological distinctions revealed as we can see in how the clusters are grouped in Figure 4D.

Figure 4- should have a simple diagram of the Pseudotime. Perhaps a model of some sort. Figure 4 legend “The point where support cells transition to progenitor cells is labeled “PC” There is no clear labeling of PC as the authors state these cells are labeled in the figure.

Authors Response: In response to Reviewer #3, we have included a cartoon schematic diagram of the pseudotime in Figure 4 as panel A. To address the unclear labeling where support cells transition to progenitor cells: We have included an arrow in Figure 4B pointing to the progenitor cell cluster that splits from support cells and included a line in the legend: “The arrow indicates the first PC cluster (Cluster 4) that is transcriptionally distinct from SCs.” We found that the aggregate of scATAC-seq profiles closely resemble bulk ATAC-seq samples, indicating that aggregate scATAC-seq captured the chromatin accessibility in a manner equivalent to bulk ATAC-seq assays (Figure 5A).

Global comment but figure is specific example – could this be addressed in text better. A hallmark of high quality ATAC-seq libraries is a banded insert size distribution with peaks or genomic signals representing putative regulatory regions resulting from nucleosome protection, which was apparent even in individual cells (Figure 5B). Reference?

Author Response: In response to “Global comment but figure is specific example – could this be addressed in text better.”: We have revised the text and include a line discussing the example presented in Figure 5A. We have also included the reference for scATAC-seq library hallmarks described in the text. “Indeed, we found a clear correlation between chromatin accessibility and nearby/distal gene expression (pvalue < 0.05, hypergeometric Test) (Figure 6D and Figure 6E; Table S15).”

Here and in other parts of this paper – direct numbers are used rather than proportions/percentages of cells. 6D is the number of DE genes that are also linked to RREs. Don't we need to see the percentage of DE genes that also intersect with a RRE? Just a number seems impossible to interpret unless you look back to the number of genes mentioned pages ago.

Author Response: We have now included percentages of differentially expressed (DE) genes that intersect with RREs in the results section. “Enhancer deletion

mutants had no overt morphological phenotypes in early larvae, and both heterozygous and homozygous deletions survived to adulthood.”

To interpret this section better, we need to know how much Sox2 is normally expressed during development and adult homeostasis. Is the enhancer regeneration specific or is still used at other times but function not critical? Figure B shows normal larvae have less hair cells to begin with – although no stats shown for that comparison.

Author Response: We agree with Reviewer #3’s comment and have provided statistical analysis for comparisons between mean hair cell number present in the lateral line neuromasts of untreated controls and untreated mutants. We examined the mean hair cells per neuromast in untreated heterozygous and homozygous enhancer deletion mutants 7 days post fertilization (dpf) and found no differences for either *sox2hg139* or *sox2hg140* either as heterozygous or homozygous deletions, and no difference for *sox2hg138* heterozygous deletions. We did find a statistically significant difference between wild-type and homozygous *sox2hg138* enhancer deletion mutants with the mutants having on average 1-2 fewer hair cells per neuromast (twoway ANOVA $p < 0.001$)(Figure S7).

“In non-regenerating sensory epithelia of the enhancer deletion mutants, *sox2* RNA levels were not significantly altered in comparison to control sensory epithelia. In adult zebrafish without the enhancer deletion undergoing hair cell regeneration, *sox2* expression was elevated on days 4, 5, and 7 post-DT. By day 9 post-DT, *sox2* expression levels were near control levels and were further reduced on day 11 post-DT (Figure 9D).” What? In non-regenerating sensory epithelia of the enhancer deletion mutants, *sox2* RNA levels were not significantly altered in comparison to control sensory epithelia. Where does this result come from?

Author Response: *sox2* mRNA levels were measured in adult (6-10 month old) sensory epithelia by quantitative real-time PCR (RT-qPCR) analysis. In non-regenerating sensory epithelia of the heterozygous and homozygous enhancer deletion mutants, *sox2* RNA levels were not significantly altered in comparison to control wild-type sensory epithelia. The data is presented in Figure 9D. We have revised the methods and main text to explain this.

“In adult zebrafish without the enhancer deletion undergoing hair cell regeneration, sox2 expression was elevated on days 4, 5, and 7 post-DT.” Figure D?

Author Response: These data were in figure 9D, they are now in figure 9E. We have changed the format of graph to show individual data points as well as to improve clarity for the reader.

Figure 9E – presumably a model but no explanation or legend. No idea what the spots represent.

Author Response: We have revised Figure 9F (9E in prior version) and its legend which includes a description of the model.

Materials and methods- “aligned to the zebrafish genome” Which zebrafish genome was used to align the data to?

Author Response: ~~We used the Danio rerio genome reference sequence danRer11~~ (GRCz11) and have now indicated the reference name and build in the methods, resources table, and text where appropriate

Referees' report, second round of review

Reviewer 1:

The authors have largely addressed my initial concerns. Most importantly, they have now confirmed the hair cell regeneration defect using multiple independent sox2 enhancer deletion alleles. This alleviates my concern of whether the original allele only affected the sox2 enhancer. This is a technical tour de force of generating important single-cell data for hair cell regeneration in the zebrafish inner ear, and impressively identifying an enhancer important for controlling precise levels of sox2 critical for timely hair cell regeneration. I just have two remaining concerns regarding the presentation/interpretation of data.

1. The description/interpretation of Fig. 8 is still inaccurate. The violin plots do not clearly show change in expression of most sox genes, contrary to what is stated in text. This is particularly true for SCs which seem to be the cell population in which sox expression would be most relevant. "The Sox genes sox2, sox4b, sox10, and sox21a all showed expression changes, as did six1a, six1b, six4a, and six4b." I think the text needs to be tempered to more accurately

describe the gene changes shown in Fig. 8, and if expression changes for each sox/six gene exist then a more unbiased statistical/quantitative test needs to be made to show that the expression changes are significant.

2. The feature plots in Fig. 5E-G remain very difficult to interpret and should be removed or visualized in a different way to highlight expression in specific clusters.

Reviewer 2:

The authors thoughtfully and carefully addressed all the reviewers concerns and in doing so made the manuscript much stronger and clearer. This manuscript presents important new findings in the field of adult hair cell regeneration and is suitable for publication in Cell Genomics.

Reviewer 3:

I am very happy with the changes and still believe this is a great paper encompassing an impressive amount of work

Authors' response to the second round of review

Reviewers' Comments:

Reviewer #1: The authors have largely addressed my initial concerns. Most importantly, they have now confirmed the hair cell regeneration defect using multiple independent sox2 enhancer deletion alleles. This alleviates my concern of whether the original allele only affected the sox2 enhancer. This is a technical tour de force of generating important single-cell data for hair cell regeneration in the zebrafish inner ear, and impressively identifying an enhancer important for controlling precise levels of sox2 critical for timely hair cell regeneration. I just have two remaining concerns regarding the presentation/interpretation of data.

1. The description/interpretation of Fig. 8 is still inaccurate. The violin plots do not clearly show change in expression of most sox genes, contrary to what is stated in text. This is particularly true for SCs which seem to be the cell population in which sox expression would be most relevant. "The Sox genes sox2, sox4b, sox10, and sox21a all showed expression changes, as did six1a, six1b, six4a, and six4b." I think the text needs to be tempered to more

accurately describe the gene changes shown in Fig. 8, and if expression changes for each sox/six gene exist then a more unbiased statistical/quantitative test needs to be made to show that the expression changes are significant.

Author Response: We agree with Reviewer # 1 that the description and interpretation of Figure 8 was incomplete. Prior to the first resubmission, we performed differential expression testing using Seurat to calculate statistically significant changes in gene expression and we failed to incorporate the information completely. The Seurat differential test using the Wilcoxon rank sum test is a widely used statistical test that can address if expression changes are significant in a cell type specific manner. We have revised the text to include p-values and fold changes. We added asterisks above cell type comparisons presented in Figure 8B to instruct the reader that those comparisons demonstrate statistical significance. The figure legend includes the p-values and fold change cutoff related to each asterisk in Figure 8B which has been moved to the Supplemental section (now Fig. S5) due to formatting guideline restrictions on figure numbers in the main text. We have revised the text which now states (page 11 line 26): “Based on the scRNA-seq data and differential (DE) expression testing performed using Seurat on sampled data, we determined which Sox and Six factors were expressed in each cell type to correlate accessible binding sites to available transcription factors (Figure S5A). In agreement with the pseudotime analysis, the sox genes were dynamically expressed (Figure S6). The Sox genes sox4a, sox4b, sox11a, and sox21a all showed differential expression changes, as did six1a, six1b, six4a and six4b (p-value < 0.01; FC \geq 0.25) (Figure S5B). Genes such as sox10 and sox11a are support cell markers. In support cells, although sox10 is virtually unaffected, sox11a (p-value < 1.64×10^{-5} ; FC = -0.77) and sox4b (p-value < 1.04×10^{-20} ; FC = -1.48) have reductions in expression during hair cell regeneration. sox4b continues to be reduced in expression in progenitor cells (p-value < 2.03×10^{-12} ; FC = -0.79) and hair cells (p-value < 1.04×10^{-20} ; FC = -1.48). The changes in sox2 and sox21a were subtle and particularly restricted to the progenitor cells according to pseudotime analysis (Figure S6) suggesting they are key drivers of the state change, while both six1 genes, but particularly six1b in both regenerating and non-regenerating sets, (p-value < 2.34×10^{-6} ; FC = 0.57), were very strongly expressed in hair cells (p-value < 0.01; FC > 0.25) (Figure S5B).”

2. The feature plots in Fig. 5E-G remain very difficult to interpret and should be removed or visualized in a different way to highlight expression in specific clusters. author response

Author Response: We agree with Reviewer #1 that the dot plots in Figure 5E-G are difficult to interpret and recognize this limitation of visualizing the scATAC-seq measurements (peaks). Visualizing activities of marker genes to help interpret scATAC-seq clusters is noisier than scRNA-seq measurements because the activities from scATAC-seq represents measurements from sparse chromatin data due to the intrinsic limitation of numbers of chromosomes per nucleus and because they assume a general correspondence between gene body/promoter accessibility and gene expression which may not always be the case (references: <https://pubmed.ncbi.nlm.nih.gov/26294014/>). We recognize this limitation and removed Figure 5E-G. We revised the text and figure legends accordingly.